# Detecting Unknowns to Predict Hallucination, Before Answering: Semantic Compression Through Instruction

## Abstract

Large language models (LLMs) excel in various tasks but often suffer from hallucinations, providing incorrect information with high confidence. To address this, we focus on detecting whether an LLM possesses enough knowledge for the answer, a concept referred to as the "feeling of knowing" (FoK). We propose a novel approach called **s**emantic **c**ompression by trying to **a**nswer in **o**ne-word (SCAO), which enables efficient FoK before generating full sentences, with only minimal computational cost. Additionally, we introduce a method to measure the self-awareness of FoK methods precisely by filtering out distracting variables, an **a**pproximate **q**uestion-dependency **e**ffect (AQE) test. Our experiments demonstrate that the feature fusion model of our SCAO and probing achieves enhanced performance in FoK in both factoid and open-ended question answering involving entity recall. The code and the dataset are available online (https://anonymous.4open.science/r/SCAO-2FF8).

## 1 Introduction

Large language models (LLMs) demonstrate impressive abilities in various applications (Ouyang et al., 2022; OpenAI, 2023). However, even the state-of-the-art LLMs often suffer from hallucination (Cohen et al., 2023). By providing just one wrong answer out of 10 correct answers, the reliability of the entire answer is undermined. While hallucination arises from various causes (failure of reading, reasoning, memory retrieval, etc), it is well-known that a major cause is when the model is asked questions on the knowledge not pre-trained (Tonmoy et al., 2024). In this sense, a simple strategy to significantly reduce hallucination is just to reject answering when it can not. Therefore, the task of determining whether an LLM knows the knowledge or not is crucial. This task has been researched recently, and we name this task as the feeling of knowing (FoK), which is a psychological term (Nelson, 1990; Koriat, 1993) referring to "The self-judgment whether a human can recall certain memory"[1]. Similarly, the FoK of LLMs can be defined as the ability to be aware of whether the model possesses specific knowledge. We introduce the term FoK to distinguish it from "hallucination detection," as the two often have been regarded as identical (Zhang et al., 2024).

Numerous outstanding works have explored FoK of LLM; however, most of them have focused on a scenario of "generating whole answer sentence and then detect" (Manakul et al., 2023; Chen et al., 2024; Ren et al., 2023; Huang et al., 2023; Kuhn et al., 2023). Though performing FoK prediction with all information may increase the accuracy, this approach is highly time-consuming and costly. This approach makes the user wait until the end of the answer to know whether it is reliable, which is less useful in a real-world service. Moreover, recent cognitive neuroscience research suggests that the human brain conducts FoK judgment in about 300 milliseconds at the unconscious level, right after being given a query (Irak et al., 2019). Additionally, this process is related to a brain region of the prefrontal cortex, which is apart from the main region for language generation (Wernicke's area) (Binder, 2015). This observation suggests that detecting whether a biological neural network holds a certain memory or not (FoK) can be partially achieved in a very short time, leveraging only the information provided before the verbalization of the answer. Moreover, one of the widely recognized

---

[1]The term FoK in psychology refers to 1) the phenomenon of "feeling like you know something but being unable to recall the name", 2) and the self-judgment of knowing. We use the term in the latter sense in this study.

theories of human FoK mechanism is the accessibility model (Koriat, 1993), which claims vividness of memory serves as an important cue.

Believing that similar efficiency is achievable in artificial neural networks, we focus on exploring the possibility of FoK before answer generation. In particular, we highlight that token-level confidence can serve as one of the key sources of information, representing the vividness of memory. Previous research has also explored token-level confidence as a method for FoK in generated text (Fadeeva et al., 2024; Lin et al., 2024), but there remains still room for further improvement. We propose a perspective that LLMs are structurally similar to dense retrievers, as both conduct maximum inner product search (MIPS) over knowledge space. As the confidence scores of retrieved documents reflect whether the queried knowledge is contained in the vector database (Zhang et al., 2022), the token-level confidence of LLM might reflect whether the queried knowledge is contained in the LLM, serving as a FoK verifier. However, this concept will make sense only when a single token embedding vector of LLM encapsulates a single piece of knowledge, just like a dense retriever. For this, the token embedding requires semantic compression, which is nontrivial. However, we discovered this could be achieved simply by using the instruction prompt "answer in only one word" without further parameter tuning. We term this method **s**emantic **c**ompression via trying to **a**nswer in **o**ne word (SCAO). We propose a method of combining SCAO with probing (Azaria & Mitchell, 2023), achieving enhanced performance in the FoK task.

In addition, for a more precise evaluation of FoK, we provide a metric that can assess whether a dataset is fit for evaluating FoK. As FoK is a fascinating topic that focuses on the LLM's self-awareness, it necessitates methods that can measure self-awareness. However, what we can explicitly measure is only whether the model answers incorrectly, which does not directly indicate the model's lack of knowledge; It can also be due to the question itself, being too ambiguous or unanswerable. To measure and filter out this question-dependent portion, we devised a novel metric without any expensive human labor, called AQE (**a**pproximately measuring the **q**uestion-dependency **e**ffect). Through the AQE test, we prove that Mintaka[2] and ParaRel OOD[3] is relatively bias-free benchmarks.

We summarize our contributions: 1) By leveraging a perspective that LLMs and retrievers are structurally similar, we develop a FoK verification method SCAO, which needs near-to-zero additional resources. 2) We propose a metric to clearly measure the model-awareness of FoK methods. 3) Experimental results in a controlled environment show that the feature fusion model of SCAO and probing achieves enhanced performance in entity recall question answering, which demonstrates their synergetic nature.

## 2 RELATED WORKS

As the human FoK has been extensively explored in cognitive psychology and neuroscience, the concepts from this field can be leveraged to structure and categorize approaches on FoK in LLM.

**FoK of Human: Cognitive Neuropsychology**    Observations of Koriat (1993); Irak et al. (2019); Brown et al. (2017) suggest that human FoK is achieved through two major processes. **1) Unconscious level:** When a query is received, in the level not directly monitored by the conscious, the brain retrieves related memories and determines whether each fits the temporal context. During this process, the orbitofrontal cortex and prefrontal cortex are activated around 300-500 milliseconds (Schnider, 2001; Irak et al., 2019), which are distinct regions from the area responsible for verbal fluency, posterior temporal lobe (i.e., Wernicke area) (Binder, 2015). Koriat (1993) suggest that the stimuli (i.e., cue for the process) for FoK include the amount of information activated, ease of access, and vividness of each memory (Koriat, 1993). **2) Conscious level:** Memories processed at the unconscious level emerge at the conscious level and are further assessed with various meta-cognitive strategies. The strategy of checking for logical and temporal consistencies between the retrieved memories is an example. FoK of humans results from the ensemble of all these underlying processes.

According to the dual-process theory of Kahneman (2011), the level of the main process can vary depending on the type of question or task. Immediate entity recall involves unconscious or implicit memory systems, while tasks that require more procedural thinking—such as solving mathematical equations or logical puzzles—engage conscious cognitive resources.

---

[2]https://github.com/amazon-science/mintaka
[3]https://github.com/yanaiela/pararel, https://github.com/shizhediao/R-Tuning

**FoK of LLM: a part of Hallucination Detection** The two verification procedures of humans roughly align with the before-generation and after-generation approaches in hallucination detection (Among this, FoK of LLM is specified to knowledge recall issue). Also, benchmarks for FoK are categorized to correspond to each process. **1) Before-generation:** Including our work, studies on the method of hallucination detection before answer generation (Mallen et al., 2022) align with the feature of the unconscious process of human FoK. Also, benchmarks with entity type question-answer (Sen et al., 2022; Elazar et al., 2021b) primarily utilize immediate memory retrieval. **2) After-generation:** Conversely, studies that assume the scenario of generating full answers one or multiple times, including methods of utilizing other tools like external retriever (Béchard & Ayala, 2024), can be aligned with the feature of conscious level (Manakul et al., 2023; Chen et al., 2024). Among benchmarks, mathematical problem solving such as MMLU (Hendrycks et al., 2021) is more closely associated with the conscious level, as it benefits more from deliberate strategies such as multi-step reasoning, rather than from vivid recall of knowledge. Additionally, answering abstract questions that require multiple steps of reasoning (benchmarks such as TruthfulQA (Lin et al., 2022), ELI5 (Fan et al., 2019a), and Natural Questions (Kwiatkowski et al., 2019)) are also more related to conscious processes.

While FoK takes a large portion of hallucinations, **not all hallucinations directly engage FoK**, which may raise confusion. For instance, hallucinations that arise in open-book tasks are more associated with issues in reading comprehension than possession of knowledge. Benchmarks mainly associated with open-book tasks are SQuAD (Rajpurkar et al., 2016), FEVER (Thorne et al., 2018).

While various tasks and methods have been grouped under the same name of "hallucination", they involve essentially different types of cognitive processes. Each process (e.g., memory retrieval, reasoning, reading comprehension) may require distinct optimal methods to address its specific challenges. Ultimately, it will be necessary to ensemble these different approaches. However, In our work, we focus on methods and benchmarks related to FoK on the knowledge retrieval question at the before-generation phase.

## 3 PRELIMINARY: CAUSAL LM IS A DENSE RETRIEVER WHEN COMPRESSED

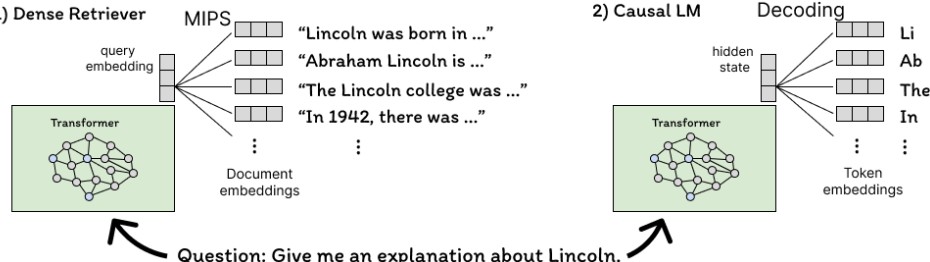

Figure 1: Structural analogy between 1) dense retriever and 2) causal LM.

The structural similarity between LLMs and dense retrievers provides an explanation for why confidence is a suitable criterion for LLM's FoK. Furthermore, this perspective provides insights for improving method with confidence score.

First, the mechanism of a dense retriever is retrieving knowledge by measuring the relevance scores or distances (e.g., inner product, cosine similarity) between the query vector and document vectors. A basic LM-powered dense retriever such as DPR (Karpukhin et al., 2020) consists of a question encoder and a vector database (DB). Embedding vector in the DB compressively represents a paragraph that contains factual knowledge. When a query is input, scores between this question vector and all the document vectors are computed. And the document vector with the highest score (or minimal distance) is retrieved, which is referred to as maximum inner product search (MIPS).

The distance can also be used to determine whether certain knowledge exists in the database. Previous research on the dense retriever system such as Faiss (Douze et al., 2024) suggests *range search*, which finds all the document vectors that are within some distance threshold. This concept can be interpreted in reverse that we can evaluate whether knowledge is within the vector DB, with a fixed

confidence threshold. For example, querying "Give me an explanation on Lincoln" to a vector DB of natural sciences may return only a few documents with confidence scores above the threshold. Querying "Give me an explanation on Newton" would likely yield more documents surpassing the threshold, indicating greater alignment between the query and the knowledge in the DB.

As an LLM is structurally analogous to a dense retriever, its confidence score can also be utilized to assess the containment of certain knowledge. Specifically, the transformer body of LLM corresponds to a question encoder, and its token vocabulary corresponds to a vector DB, if we assume each vocab represents a piece of knowledge. When the LLM inputs a query, the final layer outputs a hidden state (output embedding vector), which corresponds to the question embedding. The LM-head (decoder linear layer of the LM) conducts MIPS between this output vector and all of the token vectors, thus searching for the token with the highest confidence.

For more structural analogy, the output embedding vector should contains densely compressed information. Unlike the dense retriever, the generative LLM often infers an output vector that holds concept with low information density or focus only on the grammatical context, thus mapping to tokens with minimal semantic significance, such as "a" or "It". Therefore, we can consider that LLM becomes structurally similar to a dense retriever, only if a single output vector of LLM is **semantically compressed**: the output vector intensively aligns with vocab embeddings that contain key concept of the answer. As an intuitive example of semantic compression, for the question "What is the job of Abraham Lincoln?", an answer "president" is more semantically compressed than an answer such as "I know that Abraham Lincoln was a president", as it carries essential information in fewer expression. The semantic compression may be feasible to only a limited extent, because the vocab embedding vector is not of high density itself, containing only fragments of words (e.g., "Pr", "Ch"). Though it is nontrivial, we provide the approximate compression method in the §4.3, which empirically proves enhancement in FoK performance.

# 4 SCAO: SEMANTIC COMPRESSION BY TRYING TO ANSWER IN ONE WORD

## 4.1 TASK DEFINITION: FoK BEFORE ANSWERING

We define the FoK task as a binary classification to determine whether our target LLM $\theta$ possesses the knowledge to correctly answer a given question $q$, based on the inner state of $\theta$. In particular, this process must be completed before answer generation. To test this task, we first need to create a FoK dataset $D$, by letting the target LLM $\theta$ solve a question-answering benchmark. The benchmark should consist of question $q$ and the entity label $z$ (e.g., "Lincoln").

**FoK dataset buildup**     To build the FoK dataset, we employ the setting of Zhang et al. (2024). The $\theta$ is given $q$ to generate answer $a$ for the length of 50 tokens. It is then checked whether the entity label $z$ is contained in $a$, utilizing a string match with the normalized case. If $z$ is present in $a$, the FoK label $y$ is annotated True (or 1), otherwise as False (or 0). As a result, we get the FoK dataset $D = \{(q_1, y_1), (q_2, y_2), ..., (q_n, y_n), \}$.

**FoK task**     FoK module with learnable parameter, $\phi$, is trained on $D$ to input $q$ and predict $y$. Any method can be applied, including extracting the hidden state of $\theta$ seeing $q$ and then training $\phi$ to utilize this to predict $y$. $\phi$ can be simply a threshold or a deep neural network. In our work, we assume a scenario in which the answer token generation step of $\theta$ must be $\leq 1$.

**Real-world inference**     For a real-world scenario, we assume that when the system receives a question, it first performs a rapid FoK assessment. Based on this, the system decides whether to allow the LLM to start generating detailed answers, decline to answer, or leverage tools such as a retriever.

## 4.2 CONFIDENCE OF FIRST TOKEN IS A FoK DISCRIMINATOR

Previous works on confidence-based FoK research mostly utilize the confidence score of all tokens in the answer sentences, with normalization such as averaging (Chen et al., 2024). Utilizing more information is ultimately more advantageous; however, it also has several drawbacks. We observe a pattern that as the entity name length increases, the average confidence tends to rise. For example, Figure 2 depicts the confidence pattern of the hallucinated question-answer pair "Question: Give me an explanation about Obama. Answer: Harry Potter and the Philosopher's Stone".

Up to the token "Harry Potter", the confidence is near zero since it conflicts with the question. However, from a "philosopher", confidence increases to a near maximum, as the previous context of "Harry Potter" supports it strongly. Thus, the average confidence tends to increase regardless of whether it makes sense, when the entity name gets longer or the sentence contains more grammatical elements. This observation is supported by the analysis in Figure 3 (Left), which shows that the correlation between the mean confidence and the FoK label tends to decrease as the token increases.

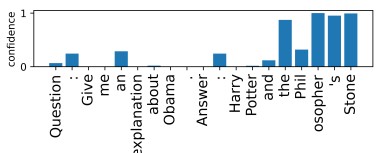

Figure 2: Probability pattern of the hallucinated answer, by LLaMA3-8B. Each bar stands for the probability (0,1) of the corresponding token.

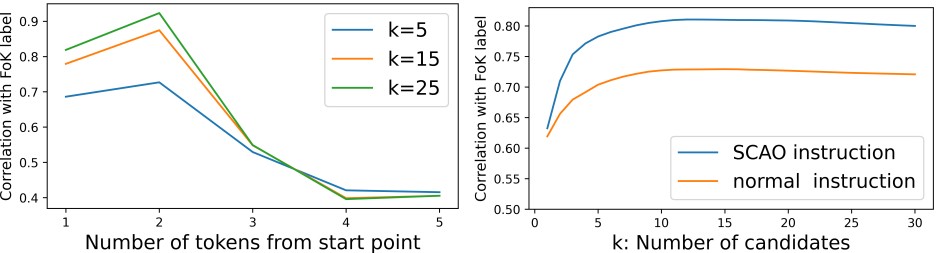

Figure 3: Y-axis is a correlation between the mean confidence and the FoK label. The X-axis of each figure stands for (**Left**) the number of tokens from the start point of the answer, and (**Right**) the number of candidates used to calculate the mean. The LLaMA3-8B and FoK datasets from Mintaka are utilized.

We also observe that averaging the confidence scores across top-$k$ vocabulary candidates, rather than just the top-1, shows a stronger correlation with the FoK label, particularly peaking around $k$=15 (Figure 3 (Right)). This suggests that incorporating more samples of distance provides more information about the relationship between the output vector and the token space.

### 4.3 SEMANTIC COMPRESSION IMPROVES FoK DISCRIMINATION

In §3, we hypothesize that when knowledge is semantically compressed into only one vector (the first token of an answer), the LLM becomes structurally more analogous to the memory retriever rather than a sentence generator, leading to the enhancement of the confidence-based FoK performance. We can achieve this compression by forcing LLM to try to answer in only one simple word. It is like guiding $\theta$ to concentrate to recall the entity, while preventing it from obscuring the point.

We can simply achieve semantic compression by querying LLM with the instruction "Answer in only one word". As described in Figure 4, we first insert $q$ into the template "[Question] {question sentence}? You must answer in only one word [Answer]" and prompt to $\theta$. The $\theta$ then infer a probability $p = p(x_j | x_{<j}; \theta)$, where $j$ is the first position of the answer, and $x_j$ represents $j$th token of the text. $P$ is a set of probability of top-$k$ vocabulary candidates $[p_1, p_2, ..., p_i, ..., p_k]$. The concept of extracting $P$, through **semantic compression with instructing to answer in one word** as described is termed SCAO.

By measuring the correlation coefficient between mean confidence and the FoK label for two instruction types (SCAO and normal), we observe that SCAO instruction shows a clearly enhanced correlation, as depicted in Figure 3.

**Approaches for discrimination of FoK label given $P$.** After the set of probabilities of top-$k$ vocab ($P$) is extracted through SCAO, it is processed by $f_\phi$ to predict between True and False. There are two approaches to process $P$: threshold-based and prediction-based.

**1) Threshold-based:** For the threshold-based discrimination, we first use the mean value of $p$ of top-$k$ vocabulary and apply the threshold, as depicted in equation 1. Here, the learnable parameter $\phi = \{\tau, k\}$ consists of a threshold ($\tau$) and the number of vocabulary candidates ($k$). During the training session, every possible pair of $k$ and threshold ($k$ is 1 to 30 in 30 steps, $\tau$ is 0 to 0.1 in 3000

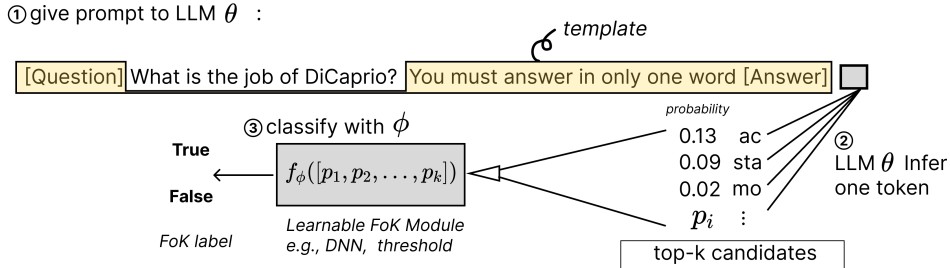

Figure 4: Pipeline of FoK through SCAO.

steps, total 90K $\{\tau, k\}$ pairs) are measured on the training dataset, and the pair with the highest accuracy is applied to the test session.

$$f_\phi(P) = \begin{cases} 1, & \text{if } \frac{1}{k}\sum_{i=1}^{k} p_i \geq \tau \\ 0, & \text{if } \frac{1}{k}\sum_{i=1}^{k} p_i < \tau \end{cases} \tag{1}$$

**2) Prediction-based:** Prediction through gradient descent is also a choice. We employ a 3-layer deep neural network (DNN) structure of input size 30 (fixed number of $k$) and output size 1 for logistic regression. The dimensions of each layer are $\mathbb{R}^{30\times40}$, $\mathbb{R}^{40\times40}$, and $\mathbb{R}^{40\times1}$. ReLU activation is applied between each layer. The objective function of DNN is binary cross entropy loss $L = -\frac{1}{N}\sum[y \cdot \log(DNN_\phi(P)) + (1-y) \cdot \log(1 - DNN_\phi(P))]$. DNN is trained on the FoK dataset while $\theta$ is frozen.

We analyze that DNN emulates the mechanism of the mean threshold approach. The weights of the first layer decide how many candidates to count in, corresponding to the function of $k$ in the threshold-based approach. The second layer decides operations, such as mean or max pooling. DNN structure is a more suitable choice if feature fusion with other data is required.

### 4.4 Feature Fusion of SCAO and Probing

Another method of utilizing the hidden state of $\theta$ is the linear probing (Li et al., 2024), which trains a linear model to predict $y$ with the input of the hidden state. As each method captivates a distinct aspect of the hidden state (as explained in Appendix B), we suggest the feature fusion of SCAO and probing. That implies utilizing both top-30 confidence value $P$ and $h_{th}$ hidden state from $\theta$ as inputs to DNN. Similar to §4.3, we employ a 3-layer DNN structure for feature fusion modeling as illustrated in Figure 5, which takes the following procedure: The $h_{th}$ hidden state from $\theta$ is first aggregated to $\mathbb{R}^{30}$ via a linear layer. Then, it is concatenated with $P$ and passed through the feed-forward network with hidden size 60.

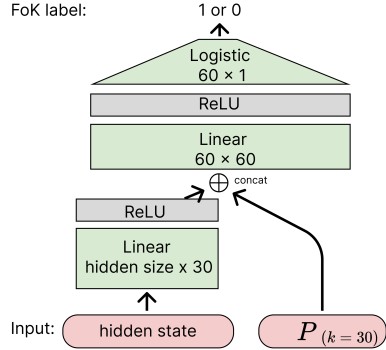

Figure 5: Structure of feature fusion with SCAO and probing.

## 5 AQE: Assessing Question-dependency Effect of FoK dataset

For a more precise evaluation of FoK, we provide a metric that can assess whether a dataset is fit for evaluating model-awareness. While FoK is the ability to be aware of whether LLM possesses specific knowledge, what we can explicitly measure is only the amount of incorrect answers (hallucinations, denote $H = \sum_{i=1}^{n} \mathbf{1}(y_i = \text{False})$), which is not enough to measure self-awareness.

Causes of incorrect answers can be naively divided into two: question-dependent and model-dependent. (1) Question-dependent: This occurs when the question is difficult or unanswerable (denote $Q$). (2) Model-dependent: This happens when the model lacks the necessary knowledge to

answer the question (denote $M$). When $H = Q + M$, FoK refers to predicting the portion of $M$. Since $H$ is measurable, we should measure $Q$ to accurately determine $M$. But measuring $Q$ directly is non-trivial, as it is for $M$.

To address this, we propose a method to approximately measure the $Q$. Our approach starts from the definition of $Q$: the portion of incorrect answers ($H$) that can be predicted solely with the properties of the question, independent of $\theta$. To fit this definition, we train and test a model $\phi$ to predict $y = \text{False}$ (incorrect answer) case using only the question as input. The closer $Q$ and $H$, the easier $\phi$ predict $y$. This accuracy of $\phi$ is the AQE score. The closer AQE is to 1, the lower the model-dependency of the dataset (paired with a certain model), making it unsuitable for measuring FoK.

We observe that the dataset with a high AQE score contains questions nearly impossible to answer correctly. We analyze this arises from the failure to properly constrain the one-to-many mapping between question and answer, which can be considered misannotation. The SimpleQA, which recorded the highest AQE (82%), contains questions like "What is a Western genre on Netflix?". Though there are countless Western genre movies on Netflix, this dataset provides only one label ("Rawhide"). Even if the LM possesses extensive knowledge about Netflix, any answer other than "Rawhide" will be labeled as incorrect. As there are multiple similar types of questions (e.g., "What is a romance genre on Netflix?", "What is a action genre on Netflix?"), this can cause the bias that any question on Netfilx is paired with only a negative FoK label. This bias makes the FoK dataset question-dependent, raising the AQE score. Even if a FoK method achieves a high score on such a dataset, it is unlikely to predict $M$ well on other datasets, or even $H$.

In contrast, Mintaka, which has a lower AQE (60%), contains questions with detailed information to ensure that each question has only one label (e.g., "Who was the first wife of Queen Elizabeth II's eldest son?"). Such questions may appear as detailed tail questions but help prevent the misannotation, resulting in lower AQE. ParaRel contains questions vulnerable to misannotation, as ParaRel ID (in-domain) shows a high AQE score. However, the separation of the out-of-domain test dataset seems to control the possibility of finding question-dependent shortcuts, resulting in a low AQE score. We describe further benchmark setting in §6.2.1

Table 1: Bias assessment on benchmarks, with the FoK dataset labeled with LLaMA3-8B model. $\text{AQE}_{acc}$ stands for AQE of accuracy, and $\text{AQE}_{auc}$ for AUROC.

(a) Accuracy: We measure 3 criteria, AQE score, $p(True)$, and $p(False)$, where $p(True)$ is the portion of True label. The lower bound for each benchmark is the maximum value among these three criteria and 0.5. The maximum value is marked as **bold**.

|  | ParaRel OOD | ParaRel ID | Mintaka | HaluEval | HotpotQA | SimpleQA |
|---|---|---|---|---|---|---|
| $\text{AQE}_{acc}$ | **55.05** | **73.65** | **60.13** | **66.68** | 66.18 | **82.36** |
| $p(True)$ | 54.14 | 54.31 | 55.01 | 37.51 | 32.71 | 19.08 |
| $p(False)$ | 45.85 | 45.68 | 44.98 | 62.48 | **67.28** | 80.19 |
| Lower bound | 55.05 | 73.65 | 60.13 | 66.68 | 67.28 | 82.36 |

(b) AUROC: The lower bound for each benchmark is the maximum value between AQE and 0.5.

|  | ParaRel OOD | ParaRel ID | Mintaka | HaluEval | HotpotQA | SimpleQA |
|---|---|---|---|---|---|---|
| $\text{AQE}_{auc}$ | 55.02 | 82.24 | 63.63 | 65.25 | 66.78 | 68.13 |

Based on the results in Table 1(a), we exclude SimpleQA(Yin et al., 2016) and ParaRel ID in this work, which shows a high $\text{AQE}_{acc}$ score. We focus on the datasets ParaRel OOD and Mintaka, which show low $\text{AQE}_{acc}$ and a more balanced True/False rate. We suggest that minimum acceptable performance (i.e., lower bound) of accuracy for $\phi$ is not random chance (0.5), but rather be defined as $\max(0.5, \text{AQE}_{acc}, p(True), p(False))$, where $p(True)$ stands for the portion of True label in the FoK test dataset, and $p(False)$ is $1 - p(True)$. This is the maximum performance that $\phi$ can achieve through hacking the dataset. Also, we suggest setting the lower bound of AUROC as $\max(0.5, \text{AQE}_{auc})$, not as just 0.5.

We measure AQE scores for accuracy and AUROC, both of which are the metrics of our work. Notation $\text{AQE}_{acc}$ and $\text{AQE}_{auc}$ stands for each. We employ sentence BERT (sBERT) (Reimers & Gurevych, 2019) as $\gamma$, and LLaMA3-8B model as $\theta$ for the AQE test.

# 6 EXPERIMENT

We first conduct the main experiment on the hallucination detection benchmarks that assume a closed-book factoid long-form question-answering scenario. Based on the assessment in §5, we choose two benchmarks that show relatively low $AQE_{acc}$: Mintaka (Sen et al., 2022) and ParaRel OOD (Elazar et al., 2021a; Zhang et al., 2024). Those contain questions that have entity labels (e.g., "Which finalist gymnast did not win first place in the 2021 Olympic games?").

Then, we conduct a further experiment to investigate if SCAO is also effective for open-ended questions (Krishna et al., 2021) (e.g., "Give me an explanation about the Harry Potter series"). We use the benchmark **Explain**, that we present to evaluate open-ended long-form question answering, and ELI5 (Fan et al., 2019b). As both experiments share the same baseline and evaluation metrics, we first describe these, followed by the benchmarks for each experiment.

## 6.1 BASELINE AND METRIC

Our baseline should predict FoK label $y$ from question $q$ without letting the target LLM $\theta$ infer more than two steps. As this scenario has not been extensively explored, there are limited methodologies available. We utilize LLaMA-3[4] (`Meta-Llama-3-8B-Instruct`), one of the most advanced generative models. Additionally, we include experimental results from the larger (`Llama-2-13b-chat`) model. Further details on the baselines are in Appendix D.2.

**Confidence-Based Methods** The baselines based on the confidence are as described in §4.2. For notation, $SCAO_{thre}$ is a threshold-based method while $SCAO_{dnn}$ is DNN-based. $SCAO_{prob}$ is the feature fusion model described in §4.4. $Somewords_{thre}$ is threshold-based, but utilizes normal instruction rather than SCAO.

**Probing** We employ the linear probing method of Li et al. (2024); Azaria & Mitchell (2023); Mallen et al. (2022). As the LLM has $H$ hidden layers, $h \in \{1, 2, ...H\}$, We train $H$ number of a FoK module $\phi_h$ corresponding to each $h_{th}$ hidden state from the first token of answer. Each $\phi_h$ is trained to input $h_{th}$ hidden state to predict $y$. Then, only one $\phi_h$ with the highest accuracy on the validation dataset is used for the test session. For notation, `Probe (Linear)` indicates that the $\phi_h$ is a linear regression, while `Probe (DNN)` indicates that $\phi_h$ is 3-layer DNN.

**R-tuning** R-tuning (Zhang et al., 2024) is a method to train LLM to tell by itself whether it knows certain knowledge in yes or no. While the original work conducted R-tuning with the data form of "question + answer + sure/unsure expression" (note as R-tuning), we also train with the form without answer "question + sure/unsure expression" (note as R-tuning ($q$ only), as we assume a before-generation FoK scenario.

**Metric: accuracy, AUROC** Our metric is measuring whether $\phi$ accurately predicts the FoK label $y$, as the FoK label indicates whether the LLM $\theta$ properly answered the question. Previous hallucination detection studies (Chen et al., 2024; Ren et al., 2023) commonly use the AUROC metric to measure this, due to the binary nature of the task. This metric applies all possible thresholds to the probability score inferred by the model. However, as a common real-world setup allows only one fixed threshold, **accuracy** better reflects the actual performance experienced by users. Thus, we mainly focus on accuracy while we still include the AUROC to keep consistency with previous works.

## 6.2 EXPERIMENT ON FACTOID QUESTION ANSWERING

### 6.2.1 BENCHMARK SETUP

Both Mintaka and ParaRel OOD consist of factoid questions (e.g., "Which actor participated in George of the Jungle but did not appear in George of the Jungle 2?"), paired with the entity label $z$ (e.g., "Brendan Fraser"). We build the FoK label $y$ dataset according to the process in the §4.1: first $\theta$ freely generate a long-form answer for a max token 50. Then, we check whether the entity label is contained in the answer using a string match (following Stelmakh et al. (2023)). We also experiment on HaluEval (Li et al., 2023) and HotpotQA (Yang et al., 2018). Details are in Appendix E.

---

[4]https://github.com/meta-llama/llama3

**Mintaka**  As Mintaka consists of five types of questions (entity, boolean, numerical, date, string), we utilize only the entity type. This is to avoid the effect of misannotation, as we find some questions of numerical and date type (e.g., "How old is the quarterback of the Tampa Bay Buccaneers?") impossible to address without further clue. Also, the boolean type (yes-no question) is too easy to guess. And we only use English questions and exclude instances with multiple labels.

**ParaRel OOD**  We utilize the rearranged version by (Zhang et al., 2024). This version consists of train, ID (in-domain), and OOD (out-of-domain) sets. The ID set contains questions with forms and categories that are shared with the training dataset, while the OOD set does not. Based on the analysis of the AQE score in §5, we only utilize the OOD set.

### 6.2.2 RESULTS

**Feature fusion of SCAO and probing ($\text{SCAO}_{probe}$) achieves the best performances.**  As illustrated in Table 2, the probe method performs well in Mintaka, while SCAO shows better performance in the ParaRel OOD. Interestingly, the feature fusion of the two methods, $\text{SCAO}_{probe}$, achieves tie or slightly higher scores comparing the top-performing models in each benchmark. This suggests that $\text{SCAO}_{probe}$ effectively combines the strengths of each individual approach. A further experiment on the larger model (LLaMA2-13B) shows similar trends (§C.2).

R-tuning was close to random and even lower than the lower bound. This supports the concepts introduced in §2 that verbal fluency and the function of FoK are less dependent.

Table 2: FoK accuracy of LLaMA3-8B, examined on two benchmarks (Mintak and Pararel OOD).

|  | ParaRel OOD | | Mintaka | | HaluEval | | HotpotQA | |
|---|---|---|---|---|---|---|---|---|
|  | **Accuracy** | **AUROC** | **Accuracy** | **AUROC** | **Accuracy** | **AUROC** | **Accuracy** | **AUROC** |
| lower bound | 55.05 | 55.02 | 60.13 | 63.63 | 66.68 | 65.25 | 67.28 | 66.78 |
| R-tuning | 60.24 | 65.84 | 47.67 | 57.31 | 38.68 | 61.00 | 35.60 | 61.70 |
| R-tuning ($q$ only) | 54.13 | 53.84 | 56.13 | 70.97 | 63.37 | 61.63 | 69.51 | 67.52 |
| Probe (Linear) | 66.91 | 75.31 | 68.95 | 75.70 | 74.24 | 79.78 | 77.69 | 82.03 |
| Probe (DNN) | 68.63 | 75.67 | 69.00 | 76.42 | 74.57 | 80.11 | 77.43 | 81.45 |
| Somewords$_{thre}$ | 58.86 | 56.22 | 64.94 | 69.61 | 71.35 | 75.98 | 71.90 | 74.80 |
| $\text{SCAO}_{thre}$ | 70.77 | 76.33 | 67.17 | 70.99 | 74.08 | 79.23 | 75.33 | 79.27 |
| $\text{SCAO}_{probe}$ | **72.05** | **77.58** | **70.53** | **77.06** | **76.23** | **82.28** | **78.06** | **83.05** |

**$\text{SCAO}_{probe}$ shows clear performance gain in Pararel OOD.**  $\text{SCAO}_{probe}$ exhibits a clear performance gain compared to the Probe especially in ParaRel OOD, while the gain is small in others. We suggest the following rationales. First, Pararel OOD focuses more on straightforward recall of certain entities (e.g., "Where was Clonaid founded?"), while other datasets like Mintaka contain questions that require more complicated processes, including multiple steps of reasoning and comparing, multiple entity recalls (e.g., "Which Quentin Tarantino movie was nominated for Best Director in 1995 but did not win?"). This observation supports the concept presented in §2, that different types of tasks (immediate entity recall vs. multi-step reasoning) require different detection approaches, and SCAO is more optimal for FoK of immediate entity recall.

Second, Pararel OOD has the lowest AQE score, leaving little room for shortcut interference. As SCAO is a method that only takes 30 confidence scores as input, it is completely isolated from granular information of the question, which limits question-dependency. This contrasts with the probe, which accesses high-dimensional space where semantic characteristics of the question are available, thus directly absorbing and enjoying the question-dependency effect. For this reason, SCAO has more advantages in the environment with a low question-dependency effect. Third, SCAO's stronger generalization ability likely contributes to its robustness in out-of-domain settings, as further explained in the Appendix B.

**SCAO instruction clearly enhances FoK performance compared to normal instruction.**
Ablation study (§C.1) suggests that semantic compression through SCAO instruction achieves clearly better performance than normal instruction, supporting our hypothesis that semantic compression improves FoK ability.

## 6.3 Experiment on Open-Ended Question Answering

### 6.3.1 Benchmark Setup

An open-ended question-answering scenario assumes there can be numerous correct answers depending on the perspective. As it is challenging to properly evaluate open-ended answers with automated metrics such as string matching or ROUGE, we employ G-eval (Liu et al., 2023). First, we let $\theta$ respond to a given question with a max token length 50, and then request GPT4o-mini API (Achiam et al., 2023) to evaluate whether the generated answer contains no factual inaccuracies with no reference label, returning a True or False judgment (with the prompt presented in Appendix F). We utilize two benchmarks, ELI5-small and Explain, that are detailed in Appendix E.2. Additionally, we exclude the R-tuning baseline as it shows poor performance in the main experiment.

**ELI5-small** ELI5 dataset comprises 270K threads from the Reddit forum "Explain Like I'm Five". We randomly sample 16K threads and split them into training, validation, and test sets for 8:2:2 ratio.

**Explain** We present a benchmark **Explain** to evaluate a model's ability to provide a descriptive answer to an open-ended question. Explain is an extended and refined version of an open-ended long-form dataset in the well-known and verified work of FActScore (Min et al., 2023). In FActScore, a small dataset is devised to test fact-checking pipeline for long-form QA. This dataset is created by appending prompts like "Tell me a bio of <entity>" to person names sourced from Wikipedia. However, its subjects are limited to only person names, and it includes only 500 entries. To address this, we developed Explain. Explain covers more general categories such as people, history, buildings, culture, etc (the entities from Mintaka), with the dataset size expanded to about 15000 entries. The prompt is "Please give me an explanation about <entity>", which follows the concept of the dataset in FActScore. We provide more details in Appendix E.2

### 6.3.2 Results

**The $\text{SCAO}_{probe}$ shows a significant performance gain on Explain, while a rare gain on ELI5.** Similar to the observation in §6.2.2, we suggest that this result arises from the difference in type of questions between the two benchmarks. ELI5 covers questions involving analysis, comparison, causality, and methods, which require complex and extended reasoning processes, which corresponds to the conscious process in human mind.

Table 3: FoK accuracy of LLaMA3-8B, examined on two benchmarks (ELI-small and Explain).

| | ELI5-small | | Explain | |
|---|---|---|---|---|
| | **Accuracy** | **AUROC** | **Accuracy** | **AUROC** |
| lower bound | 59.88 | 61.96 | 67.75 | 72.90 |
| Probe (Linear) | 66.76 | **72.03** | 76.06 | 83.40 |
| Probe (DNN) | 66.37 | 71.48 | 76.90 | 84.56 |
| Somewords$_{thre}$ | 56.85 | 53.83 | 55.69 | 46.24 |
| SCAO$_{thre}$ | 57.62 | 53.47 | 59.60 | 61.85 |
| SCAO$_{probe}$ | **66.92** | 71.99 | **80.07** | **87.33** |

One of the questions in ELI5 ("Running, sprinting, and jogging. What's the difference?") can be an example. For a human to answer this question, we must first retrieve information on the three entities "running", "sprinting", and "jogging". Then, we start comparing the detailed features of each entity and list the similarities and differences. It already takes a few seconds for us to go under such process. On the other hand, Explain consists of questions that are focused to retrieve information relevant to the entity (e.g., "Please give me an explanation on Usain Bolt").

This analysis suggests that the SCAO is the FoK method that is optimal for questions requiring retrieval of information on a specific entity, regardless of whether the question is factoid or open-ended. Again, this result supports the concept presented in §2: the term "hallucination" actually consists of various subtypes, and each subtype requires each distinct optimal approach.

Moreover, the experiments show that the feature fusion ($\text{SCAO}_{probe}$) consistently shows the best performance, by selecting only the best outcomes from both the SCAO and Probe methods. This also supports our concept, that the ultimate solution for hallucination will be an ensemble of optimal methods for distinct subtypes.

Further analysis and the experiment on longer response length are in Appendix C.3. And As an open-ended question (such as Explain) is non-trivial to answer by a single word, we provide analysis on the reacting pattern of SCAO on questions from Explain.

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

## A    CONCLUSION AND LIMITATION

By leveraging the concept that LLMs and retrievers are structurally analogous, we suggest a hypothesis that semantic compression enhances the utility of confidence of LLM for the FoK task. And we demonstrate it through extensive experiments. Particularly, SCAO exhibited a clear performance improvement in straightforward entity recall tasks, while the gains were relatively modest for questions requiring multiple reasoning steps. Additionally, there remain limitations and future research subjects on the following topics.

**Semantic compression of vocab embedding vectors.**    As described in §3, the output embedding vector is semantically compressed, while the vocab embedding vector is still of low density. This property might limit the extent of compression, resulting in limited performance gain. If we find a way to compress the vocab embedding vector to represent knowledge more densely, we may anticipate further improvement in FoK performance.

**FoK on the temporally evolving knowledge.**    SCAO relies on LLM's confidence to determine FoK, making it hard to handle temporally evolving knowledge. This is a common issue of all approaches that utilize LLM's inner state, including probing. We can gain insights from the neuro-cognitive domain. According to Gündoğar & Demirci (2007), humans store time-related information alongside knowledge memories. When retrieving memories, the temporal relevance of the information is unconsciously evaluated. A similar idea is proposed in the dense retriever literature (Liska et al., 2022), where document embeddings are encoded with temporal metadata and retrieved with consideration of temporal context. Applying this concept, if a system is developed where LLMs store information with temporal grounding, the confidence score could reflect the temporal relevance of the information. Our work is significant as it contributes to this ultimate solution.

**Comparison with full sentence generation scenario.**    We assume that after-generation approaches are provided with more information, thus yielding better FoK performances. However, extensive experimentation is required to investigate the performance gap.

## B    EXPLANATION ON THE SCAO AND PROBE

In the main experiment (§6.2.2), SCAO outperforms the probe with a larger gap in OOD settings, indicating the robust generalization ability of SCAO. We suggest the following rationale for this result.

SCAO and probing are fundamentally similar. Probing directly utilizes the raw $h_{th}$ hidden state of $\theta$, while SCAO focuses on the last hidden state of $\theta$, which is projected onto the vocab embedding space.

Let us assume a knowledge space ($S_k$) (Figure 6), which represents the embedding of each knowledge in the $\theta$. And we term the gray area in the $S_k$ as a **boundary of knowing** of $\theta$, which represents the area where $y = 1$. This space is hypothetical and unknown but needs to be discovered to perform a FoK task for $\theta$. What we have at hand are 1) the 4096-dimensional (in the case of LLaMA3-8B) hidden states ($S_h$) and 2) a vocab embedding space ($S_e$) of the same dimension, with vocab embedding vectors ($v$) distributed across $S_e$. In probing, a linear layer is trained to map $S_h$ to $S_k$. The weight of the linear layer is supposed to be a direction vector that represents a principal component of the boundary of knowing. Thus, an inner product with this vector tells if the given hidden states match the direction. Since it utilizes all 4096 dimensions to describe $S_k$, it offers high informational resolution, leading to generally strong performance.

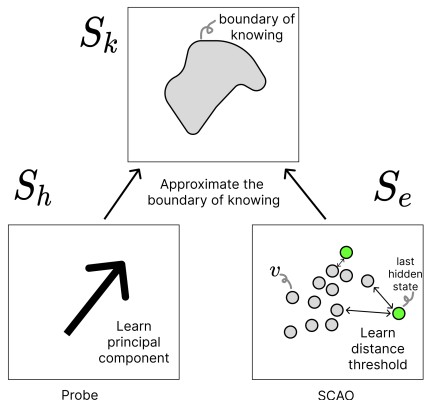

Figure 6: Illustration on two methods (probe, SCAO) approximating the boundary of knowing of $\theta$. In $S_e$ (lower right), the green balls are the last hidden state vector that is mapped to the vocab space. SCAO learns the threshold of distance between the hidden state and $v$ to classify $y$ of each ball.

Conversely, SCAO assumes that $S_e$ approximately aligns with $S_k$ and the $v$ aligns with the boundary of knowing when the key (the last hidden state $h_l$) is semantically compressed. SCAO figures the shape of $S_k$ by measuring the distance between $h_l$ and other samples $v$ in $S_e$. These mechanisms yields the following properties: 1) SCAO leverages $S_e$, thus utilizing more information than probing. 2) However, this information is compressed into a single scalar value, distance, leading to lower information resolution, showing lower performance than the probe. 3) Despite the lower resolution, this simplification appears to enhance generalization. For instance, in out-of-domain scenarios, probing struggles with unfamiliar features in $S_h$, while SCAO effectively handles these novel features by employing its simplified distance-based measure.

Since probing and SCAO reflect slightly different aspects of $S_k$, combining these two methods in a feature fusion appears to provide an additional performance boost by leveraging their complementary strengths.

## C ADDITIONAL EXPERIMENTS

### C.1 ABLATION

Table 4: FoK accuracy of LLaMA3-8B, examined on two benchmarks (Mintak and Pararel OOD). The detailed setting for benchmark and baselines are in §6. The best performance is marked as **bold** while the second best is underlined.

|  | Mintaka | | ParaRel OOD | |
| --- | --- | --- | --- | --- |
|  | **Accuracy** | **AUROC** | **Accuracy** | **AUROC** |
| *somewords (instruction) thre acc* | 64.94 | 69.61 | 58.86 | 56.22 |
| *somewords (instruction) thre corr* | 64.94 | 69.61 | 58.77 | 56.32 |
| *oneword (instruction) thre acc* | **66.76** | 70.79 | **70.78** | 76.33 |
| *oneword (instruction) thre corr* | 66.34 | 70.99 | 62.04 | **76.79** |
| *oneword (instruction) DNN* | 65.38 | 70.37 | 70.55 | 76.28 |
| *oneword (finetune) thre acc* | 66.15 | **72.78** | 70.24 | 74.72 |
| *oneword (finetune) DNN* | 66.03 | 71.74 | 69.38 | 75.06 |

In this part, we examine variations of extracting and utilizing confidence from $\theta$. For the notation of Table 4, *somewords* indicates normal instruction, while *oneword* indicates SCAO manner. The expression *thre* indicates that the model employs threshold and $k$ as a $\phi$. Suffix *acc* indicates *corr* indicates that the optimal threshold is chosen according to the accuracy, and correlation coefficient, respectively. Baseline with *finetune* in the name modifies $\theta$ with fine-tuning LoRA adapter to answer in one word. The detailed method is in §D.2.1). In this manner, to confirm, *oneword (instruction) thre acc* corresponds to $\text{SCAO}_{thre}$, and *somewords (instruction) thre acc* corresponds to $\text{Somewords}_{thre}$ in Table 2. The observation of the experiment is as follows.

**Semantic compression shows clear improvement in FoK.** The accuracy of *oneword* instruction-based methods shows better performance than *somewords*-based, as *oneword (instruction) thres acc* gets 70.78% and *somewords (instruction) thre acc* gets 58.86% accuracy in ParaRel. This indicates that the semantic compression clearly enhances the utility of confidence value for FoK.

**Instruction is enough for compression.** The accuracy of *oneword (instruction) thre acc* is similar or even better than *oneword (finetune) thres acc* in both benchmarks. This demonstrates that sufficient compression can be achieved purely through instruction, leveraging the reading comprehension and reasoning ability of LLM. Additionally, we observe that fine-tuning causes the forgetting of knowledge during training the LoRA adapter. Consequently, a gap arises between the FoK label and the real amount of knowledge stored. This mismatch could cause a distraction to $\phi$.

**Threshold is better than DNN.** *oneword (instruction) thre acc* shows better performance than *oneword (instruction) DNN*. As a rationale for this phenomenon,

## C.2 EXPERIMENT ON LARGE MODEL

We conduct the main experiment with the larger (LLaMA2-13B) model. It shows consistent trends with the §6. The Table 5 describes experiments on the FoK datasets, and Table 6 describes the AQE scores.

Table 5: FoK assessment of LLaMA2-13B with factoid and open-ended question-answering benchmarks.

|  | PraRel OOD | | Mintaka | | HaluEval | | HotpotQA | |
|---|---|---|---|---|---|---|---|---|
|  | **Accuracy** | **AUROC** | **Accuracy** | **AUROC** | **Accuracy** | **AUROC** | **Accuracy** | **AUROC** |
| lower bound | 53.49 | 56.80 | 60.78 | 57.95 | 68.85 | 63.81 | 73.23 | 67.06 |
| R-tuning | 55.11 | 57.56 | 54.83 | 48.38 | 38.39 | 53.29 | 32.72 | 52.13 |
| R-tuning ($q$ only) | 56.12 | 48.15 | 54.55 | 52.22 | 38.51 | 54.41 | 33.05 | 55.54 |
| Probe (Linear) | 68.20 | 74.98 | 68.63 | 75.46 | 76.20 | 79.52 | 76.85 | 79.72 |
| Probe (DNN) | 68.76 | 73.82 | 69.51 | 76.45 | 76.55 | 81.01 | 77.21 | **80.87** |
| Somewords$_{thre}$ | 56.35 | 60.47 | 59.50 | 63.28 | 67.75 | 64.53 | 67.75 | 64.53 |
| SCAO$_{thre}$ | 64.43 | 72.96 | 65.69 | 70.83 | 73.05 | 75.89 | 73.05 | 75.89 |
| SCAO$_{probe}$ | **72.45** | **79.06** | **70.07** | **77.01** | **78.00** | **82.91** | **78.24** | 80.87 |

Table 6: AQE scores of the FoK dataset labeled with LLaMA2-13B model. AQE$_{acc}$ stands for AQE of accuracy, and AQE$_{auc}$ for AUROC.

(a) Accuracy: We measure 3 criteria, AQE score, $p(True)$, and $p(False)$. The lower bound for each benchmark is the maximum value among these three criteria and 0.5. The maximum value is marked as **bold**.

|  | ParaRel OOD | Mintaka | HaluEval | HotpotQA |
|---|---|---|---|---|
| AQE$_{acc}$ | 52.86 | **60.78** | **68.85** | **73.23** |
| $p(True)$ | 46.50 | 49.58 | 33.27 | 29.79 |
| $p(False)$ | **53.50** | 50.42 | 66.73 | 70.21 |
| Lower bound | 53.50 | 60.78 | 68.85 | 73.23 |

(b) AUROC: The lower bound for each benchmark is maximum value between AQE and 0.5.

|  | ParaRel OOD | Mintaka | HaluEval | HotpotQA |
|---|---|---|---|---|
| AQE$_{auc}$ | 56.80 | 57.95 | 63.81 | 67.06 |

## C.3 OPEN-ENDED QUESTION ANSWERING

We present additional experimental results with a longer max token length (256), specifically focusing on comparisons with the key baselines (Table 7). As the response length increased, the average FoK accuracy of ELI5 improved. However, the overall tendency of the FoK results remains the same: SCAO clearly outperforms in Explain, while there is no significant difference in ELI5.

Table 7: FoK accuracy of LLaMA3-8B, examined on two benchmarks (ELI-small and Explain), with max token length 256.

|  | ELI5-small | | Explain | |
|---|---|---|---|---|
|  | **Accuracy** | **AUROC** | **Accuracy** | **AUROC** |
| lower bound | 67.19 | 67.02 | 75.37 | 74.77 |
| Probe (Linear) | 75.00 | **79.84** | 77.13 | 82.63 |
| Probe (DNN) | 74.65 | 78.98 | 77.98 | 83.87 |
| somewords$_{thre}$ | 65.23 | 51.50 | 67.63 | 42.78 |
| SCAO$_{thre}$ | 64.87 | 55.21 | 67.92 | 60.51 |
| SCAO$_{probe}$ | **75.03** | 79.29 | **80.61** | **86.72** |

**The accuracy of SCAO$_{thre}$ is below the lower bound.** On the Explain benchmark with max token length 50 (Table 3), we find that while SCAO$_{probe}$ performs best, SCAO$_{thre}$ falls below the lower bound, which is counter intuitive. We suggest following rationale for this phenomenon. As SCAO is isolated from the question-awareness, we should consider that the lower bound for the SCAO approach shall not include AQE$_{acc}$, which is 55.76 for Explain. In this perspective, the accuracy of SCAO$_{thre}$ (59.60) has 3.84p gain from its lower bound. This additional information seems to be aggregated with `Probe (DNN)`, resulting in a performance gain of SCAO$_{probe}$.

### C.4 HOW SCAO REACTS TO OPEN-ENDED QUESTION

As an open-ended question (such as Explain) is non-trivial to answer by a single word, it will be valuable to take a look at how model react at the first token of answer in both one-word prompt and the normal prompt. (Figure 7)

First, in non-compressed cases (queried with a normal prompt), the following patterns are frequently observed: (1) The response often starts by repeating the entity name mentioned in the query. (2) The response begins with grammatical function words such as "The" or "A". In other words, the model tends to take the easy path. As a result, the probability of the initial token is generally inflated, regardless of whether the model truly knows the subject.

On the other hand, when prompted to answer with a one-word response, the first token often corresponds to the initial token of a word encapsulating the entity's characteristics. For example, in response to the question "Please give me an explanation about 'Breaking Dawn'.", the first candidate token was "Tw" (the first token of "Twilight"). In other words, with one-word prompting, the model shows a stronger tendency to retrieve its own knowledge related to the entity. This trend is also reflected statistically. Among the 2152 test samples in the Explain dataset, the case that the top-1 candidate of the first token of the response being a component of the entity is 84.5% for normal prompting, significantly outpacing the 12.1% for one-word prompting. Similarly, the first token being "the" occurred in 17.8% of normal prompting cases, compared to just 0.02% for one-word prompting. (Table 8)

Table 8: The number and portion of each case, when questions from the test set (total 2152) of Explain are asked to the LLaMA3-8B model using various prompts. The columns represent each prompt style. In the rows, "repeating subject" refers to cases where the top-1 candidate for the first token of the answer is a component of the queried subject entity. "The" refers to cases where the top-1 token is "the."

|  | one-word prompt | normal prompt |
| --- | --- | --- |
| repeating subject | 1819 (84.5%) | 261 (12.1%) |
| "the" | 383 (17.7%) | 5 (0.2%) |

Figure 7: Y-axis is the top-7 candidates of the first token of the answer to the question "Please give me an explanation about **Breaking Dawn**". The X-axis is the probability for each candidate. **Left** is for one-word prompt, and the **Right** is for normal prompt. With the one-word prompt, the model appears to attempt to retrieve knowledge related to "Twilight," which is the series name of Breaking Dawn. In contrast, with the normal prompt, the model tends to repeat the question entity, "Breaking Down". Since it chooses the easier path, the probabilities are higher.

## D  Experiment Setup Detail

### D.1  Experiment Pipeline

First the dataset is divided into $D_{train}$, $D_{valid}$, and $D_{test}$. We fit $\phi$ to $D_{train}$, while $\theta$ is frozen. The next step varies among two types.

**Learning-Based**  The methods that need machine learning, such as $\text{DNN}_{oneword}^{inst}$ and probe, are trained on the $D_{test}$ with the objective of BCELoss. We train for five epochs and choose the checkpoint with the best accuracy on the $D_{valid}$, which yields $\phi'$. Then we use this $\phi'$ to test on the $D_{test}$. We calculate two metrics of accuracy and AUROC. When training, the learning rate is 1e-3, and the optimizer is AdamW.

**Threshold-Based**  The threshold-based methods such as $\text{Thres}_{oneword}^{inst}$ find its $\phi$ (threshold, $k$) in $D_{train}$, without evaluation on $D_{valid}$. We select the $\phi'$ (e.g., threshold, $k$) that achieves maximum accuracy by performing a search over all possible threshold values between 0 and 1 and $k$ of 1 to 30. And use this $\phi'$ to test on the $D_{test}$. AUROC is measured only with $k_{\phi'}$, without threshold$_{\phi'}$.

### D.2  Baselines detail

#### D.2.1  SCAO with finetuning

In this part, we describe the finetuning method to achieve SCAO, training $\theta$ with a focused answering pattern. The process involves the following step: (1) We build a dataset $D_{SCAO}$ with a random 0.5 portion of the training dataset that consists of a sentence with the form of "`question + instruction + one-word answer`". The one-word answer is the entity label, with the grammatical prefix (e.g., "The ") removed. For example, "[Question]:What is the job of Lincoln? Answer in only one word. [Answer]:President.". (2) We attach a LoRA adapter (Hu et al., 2021) $\pi$ to $\theta$, train $\pi$ on the $D_{SCAO}$. (3) On the inference time, let $\pi$ infer $P$ with SCAO instruction again. $\pi$ is an adapter just for performing FoK, and the real answer exhibited on the service is generated by only $\theta$.

#### D.2.2  R-tuning

Distinct from the original work, we train a separate LoRA adapter as a $\phi$, then let $\phi$ predict the FoK label of the body LLM $\theta$. The original work directly trains $\theta$ itself as a $\phi$. This modification is to address the catastrophic forgetting problem (Jang et al., 2021) during training $\theta$ directly. We observe that the True rate decreases by 13.2%p after R-tuning, seriously undermining the justification of the method. We train for one epoch with a global batch size of 16 and a learning rate of 1e-5, as it is reported that a small batch size is better for R-tuning.

## E  Benchmark Detail

**FoK labeling detail**  On the factoid question answering experiment, we build FoK labeled dataset following and modifying Zhang et al. (2024). Specifically, we raise the max token length from 5 of previous work to 50, as we observed several cases where LLM generates correct answers in a descriptive style. This modification increases the True rate of the model and benchmark pair from 38.38 to 55.01 (Mintaka with LLaMA3-8B), significantly correcting the misannotation.

### E.1  Dataset detail

In this paragraph, we present details about the benchmark dataset HaluEval (Li et al., 2023) and HotpotQA (Yang et al., 2018).

**HaluEval**  HaluEval is a dataset containing question-answering, summarization, dialogue, and user-query with correct answers and hallucinated answers. We only use the question-answering part, following (Zhang et al., 2024). An example of the question is "The Oberoi family is part of a hotel company that has a head office in what city?", paired with the label "Delhi".

**HotpotQA** HotpotQA is a question-answering dataset where each instance consists of a question, label (types including entity, boolean, numerical), and reference documents. We utilize only the question and answer to fit the closed-book scenario. An example of the question is "What government position was held by the woman who portrayed Corliss Archer in the film Kiss and Tell?", paired with the label "Chief of Protocol". We use the development dataset as a test set, following (Zhang et al., 2024).

## E.2 EXPLAIN

We present examples of questions, categories, statistics on Explain (§E.2, Table 10, Table 12)

Table 9: Examples of questions in Explain

| Questions | Entity |
|---|---|
| Please give me an explanation about "A Game of Thrones". | A Game of Thrones |
| Please give me an explanation about "Simone Biles". | Simone Biles |
| Please give me an explanation about "Winston Churchill". | Winston Churchill |
| Please give me an explanation about "Fyodor Dostoevsky". | Fyodor Dostoevsky |
| Please give me an explanation about "District 12". | District 12 |
| Please give me an explanation about "The Battle of Gettysburg". | The Battle of Gettysburg |

Table 10: #Data for the entity categories in Explain

|  | Train | Dev | Test |
|---|---|---|---|
| Music | 914 | 139 | 273 |
| History | 1059 | 149 | 296 |
| Geography | 1033 | 144 | 306 |
| Politics | 1036 | 143 | 300 |
| Video games | 1057 | 150 | 302 |
| Movies | 953 | 138 | 269 |
| Books | 1020 | 140 | 283 |
| Sports | 909 | 128 | 245 |

Table 11: Bias assessment on benchmarks, with the FoK dataset labeled with LLaMA3-8B model (both max length of answer 50 and 256). "ELI-small (50)" stands for the case with a max length of 50. As the answer length increases, the likelihood of errors rises, and the true rate tends to decrease. $AQE_{acc}$ stands for AQE of accuracy, and $AQE_{auc}$ for AUROC.

(a) Accuracy: We measure 3 criteria, AQE score, $p(True)$, and $p(False)$. The lower bound for each benchmark is the maximum value among these three criteria and 0.5. The maximum value is marked as **bold**.

|  | ELI5-small (50) | Explain (50) | ELI5-small (256) | Explain (256) |
|---|---|---|---|---|
| $AQE_{acc}$ | **59.88** | **67.75** | **67.19** | **75.37** |
| $p(True)$ | 57.71 | 44.24 | 35.85 | 32.58 |
| $p(False)$ | 42.29 | 55.76 | 64.14 | 67.41 |
| Lower bound | 59.88 | 67.75 | 67.19 | 75.37 |

(b) AUROC: The lower bound for each benchmark is maximum value between AQE and 0.5.

|  | ELI5-small (50) | Explain (5) | ELI5-small (256) | Explain (256) |
|---|---|---|---|---|
| $AQE_{auc}$ | 61.96 | 72.90 | 64.14 | 74.77 |

### E.3 DATA STATISTICS

We present data statistics of our main benchmarks, Mintaka and ParaRel OOD. And we also utilize ELI5-small and Explain benchmarks for long form question experiments. The number in Table 12 is the final version after filtering and preprocessing. For the ParaRel OOD, we utilize ParaRel ID as a validation dataset.

Table 12: #data in each benchmarks

|       | ParaRel OOD | Mintaka | HaluEval | HotpotQA | ELI5-small | Explain |
|-------|-------------|---------|----------|----------|------------|---------|
| Train | 5575        | 7583    | 6000     | 8000     | 9838       | 7583    |
| Valid | 5584        | 1075    | 2000     | 2000     | 3280       | 1075    |
| Test  | 13974       | 2152    | 2000     | 7405     | 3280       | 2152    |

## F    INSTRUCTION PROMPTS

In this section, we compile the instructional prompts employed in our study. Terms marked with `underline` indicate placeholders that need to be filled with the corresponding content.

A. Normal instruction template

> **[Question]:{question} [Answer]:**

B. SCAO instruction template

> **[Question]: {question} You must answer in only one word. [Answer]:**

C. G-eval instruction template

> **[instruction] The text provided within the triple backticks (''' ''') is a Question and an Answer by an agent. Your task is to evaluate whether the agent's response is factually correct or incorrect.**
>
> **1) Very briefly and shortly explain whether the answer contains any factual inaccuracies.**
> **2) Finally, classify the answer as either "True" (factually correct) or "False" (factually incorrect).**
>
> **'''**
> **[Question]:{question} [Answer]:{answer}**
> **'''**

## G    FURTHER RELATED WORKS

**Semantic Compression**    Compression of LLMs is currently explored, with most studies focusing on scenarios where a large knowledge is compressed into few or a single embedding vector (Ge et al., 2024; Cheng et al., 2024). This vector is not projected into the token space but is directly fed into the LLM, functioning like an externally supplied hidden state vector. This vector is utilized as a replacement for real text documents in a scenario of retriever-augmented generation.

Our method shares a similar concept of compression, but assumes distinct scenarios and pipelines. In our work, the compressed vector is projected to the token embedding space, and the distances between the compressed vector and token embedding vectors become a key variable. This distance is utilized as a measure of FoK.

