# OpenReview forum: "Detecting Hallucination Before Answering: Semantic Compression Through Instruction"
_ICLR.cc/2025/Conference — Submitted to ICLR 2025_

### Official Review · Reviewer_qsGQ · 2024-10-31

**Soundness:** 2
**Presentation:** 2
**Contribution:** 3
**Rating:** 5
**Confidence:** 4

**Summary:**

The paper presents a novel approach to detect when a large language model (LLM) knows or doesn't know an answer, a concept referred to as the "feeling of knowing" (FoK). The authors propose Semantic Compression by trying to Answer in One-word (SCAO), an efficient method for FoK detection that requires minimal computational resources. The paper also introduces a method to measure the impact of confounding variables in benchmarks, called the Approximate Misannotation Effect (AME) test. The experiments show that combining SCAO and probing improves FoK detection in both short and long-form entity questions. The paper argues that LLMs are structurally similar to dense retrievers and that token-level confidence can be a key source of information representing the vividness of memory. It also highlights the issue of misannotation in previous research and proposes an AME test to measure it. The results show enhanced performance of the feature fusion model of SCAO and probing, demonstrating their synergistic nature.

**Strengths:**

1) The authors propose a novel approach, Semantic Compression by trying to Answer in One-word (SCAO), to detect when a Large Language Model (LLM) knows or doesn't know an answer. The paper also introduces the Approximate Misannotation Effect (AME) test, a new method to measure confounding variable effects in benchmarks.
2) By detecting when an LLM knows or doesn't know an answer before generating a full sentence, the SCAO method could significantly reduce computational cost and improve user experience. The AME test could also help researchers better understand and control the effect of misannotation in benchmarks.

**Weaknesses:**

The paper has some foundamental flaws as outlined. Nonetheless, after addressing these problems, it could potentially be a captivating piece of work.
1) The problem that the paper is attempting to solve is not well justified. The concept of "Detecting hallucination before answering" appears contradictory for two primary reasons. Firstly, the term "detecting" implies that an answer already exists to be detected, which is inconsistent with the concept of doing so prior to providing an answer. From my perspective, it appears more akin to "predicting hallucination before answering". Secondly, the necessity for "predicting" before answering is not adequately explained. Although there may be valid reasons for this, the paper fails to provide them.
2) The paper conflates the concept of "feeling of knowing" with actual "hallucination". The authors assert that their focus is on determining when a Large Language Model (LLM) knows or does not know an answer, but they do not reference or compare their work to existing studies on the subject, such as the well-regarded "Do Large Language Models Know What They Don't Know? (Z Yin, 2023)". Furthermore, while the "feeling of knowing" is related to hallucination, the two are not identical.
3) The experimental design is biased. The paper specifies that "Our baseline should predict FoK label y from question q without allowing the target LLM \theta to infer more than two steps". However, this limitation is based on the paper's specific methodology, not the task itself, and as such is unfair to the baselines.
4) The research on the impact of misannotation does not have a significant link with the SCAO, suggesting it could be divided into two separate papers.

**Questions:**

LN013: "we focus on detecting when an LLM knows or does not know an answer", feeling of knowing is a different task from hallucination detection.

LN068: "the effect of misannotation", what specific effect does it refer to?

LN080: I cannot see the connections between the contribution 1) and 2).

LN105: "FoK of LLM" misses the studies about "know and don't know in LLMs". For example, "Do Large Language Models Know What They Don't Know? (Z Yin, 2023)"

LN312: "misannotation effect" need literature support or justification. Why is it a problem and why is it important for your topic?

LN388: "Our baseline should predict FoK label y from question q without letting the target LLM \theta infer more than two steps", why should the baseline follow the restriction of your method?

LN429: "impossible to address without further clue", what does it mean? Need further explanation.

LN430: "we only use English questions and exclude instances with multiple labels", why?

---

> ### Author Response · Authors · 2024-11-18
> **Response to Reviewer qsGQ [1/3]**
>
> First of all, we appreciate your constructive feedback on our work. However, it seems there might be some misunderstandings about our work. Most of the queries raised are addressed in Sections 1 and 2 of our paper; we suggest reexamining these sections for clarification.
>
> As you mentioned, “Hallucination detection” and “FoK” are distinct concepts, and it seems misunderstandings stem from confusion between these two. Though we emphasized multiple times in Sections 1 and 2 that these two concepts are different, it seems there is still potential for misreading. We will revise the paper to eliminate any potential confusion.
>
> We address the key issue raised by the reviewer in the comments below:
>
> ---
>
> **1. “Hallucination detection” is not identical with FoK**
>
> Here, we review the definitions of "hallucination detection" and "FoK". While hallucination indicates general incorrect answering, its causes can be categorized into various factors (e.g., failure of reasoning, failure of reading comprehension, lack of knowledge). Among these, "Feeling of Knowing" refers to detecting the lack of certain knowledge. Therefore, FoK can be considered a subset of hallucination detection. Further, our work focuses on the scenario of detecting before generation.
>
> Despite these differences, we used "hallucination detection" in the title for the following reasons: 1) In common service environment like chatGPT, it is well-known that a major cause is when the model is asked questions in subjects where it lacks pre-trained knowledge. LLMs lack the ability to say "I don't know" when they encounter unknowns, leading to a hallucinated answer instead. 2) Previous works have often regarded these two concepts as identical [2], without providing a separate definition of FoK. In connection with prior research, we chose this expression.
>
> However, this can cause confusion. Thus, to avoid confusion, we thoroughly emphasized the distinction between the two concepts in Sections 1 and 2. Nevertheless, as there is still potential for misunderstanding, we revised the paper and changed the title to “Detecting Unknowns to Predict Hallucination, Before Answering: Semantic Compression Through Instruction”.
>
> ---
>
> **2. Why AME is crucial in this work**
>
> The confusion between "hallucination detection" and "FoK" seems to have prompted these questions about the importance of AME. AME is essential for measuring the performance of FoK methods (not just for SCAO), though it has not been adequately addressed in previous work.
> FoK is the ability to be aware of whether LLM possesses specific knowledge. To evaluate this, what we can explicitly measure is the amount of incorrect answers (hallucinations, denote $H$). Causes of incorrect answers in closed-book scenarios can be naively divided into two: question-dependent and model-dependent. **(1) Question-dependent:** This occurs when the question is difficult or unanswerable (denote $Q$). **(2) Model-dependent:** This happens when the model lacks the necessary knowledge to answer the question (denote $M$).
>
> When $H=Q+M$, FoK refers to predicting the portion of $M$. Since H is measurable, we should measure $Q$ to accurately determine $M$. Measuring $Q$ is exactly what AME does, as it predicts H based solely on question-dependent factors. As our work focuses on the model-aware side (FoK) and compares the FoK performance of methods, we should filter out the portion of $Q$.
>
> (To emphasize this aspect, we redefine AME as **AQE (approximate question-dependency effect)** in the revised paper)

---

> ### Author Response · Authors · 2024-11-18
> **Response to Reviewer qsGQ [2/3]**
>
> **3. Why is FoK “before answering” important**
>
> The primary motivation for FoK before generation is efficiency. In real-world services, generating a response often exceeds 1000 autoregressive steps, which takes waiting. If we can predict the reliability of the answer in just one step instead of 1000, the efficiency gain is substantial. This motivation was explained in the first section of our paper. As it is easily acceptable, we did not repeat it later.
>
> Despite its importance, this scenario remains underexplored, as most studies focus on the scenario of detecting hallucinations after generating one or more versions of answers. This is why it has become a focus of our research, and why we could not compare many baselines.
>
> ---
>
> **4. How our experiment design is derived**
>
> We identified "FoK before answer generation" as a critical task for efficiently preventing hallucination. We derived our experimental design from this task scenario. In this scenario, the FoK label y must be predicted before answer generation. At the same time, as the FoK is being aware of the model’s inner state, it needs at least one of model inference step. This is indicated in the description provided in the paper: *"Our baseline should predict the FoK label y from the question q without allowing the target LLM to infer more than two steps."*
>
> We developed our method to align with this task scenario and gathered appropriate baselines for comparison. However, despite the importance, this scenario has been underexplored. As a result, there were few existing baselines for direct comparison. So we could not compare many baselines, and had to modify some answer generation methods to fit the scenario, to our best.
>
> ---
>
> **5. Why we did not cite [1]**
>
> Thank you for introducing such an inspiring paper, “Do LLM Know What They Don’t Know?”[1]. However, we also thoroughly reviewed the paper and concluded that it addresses a different subject from ours. Among the causes of the hallucination, [1] focuses on the question-dependent side, while we focus on the model-dependent side.
>
> While this work addresses a different subject from ours, yet this work uses a similar terms, which can cause confusion to our readers. It seems the reason you think [1] addresses the same subject as ours is also due to this confusion.
>
> We explain the terms of [1] here. [1] define the term “self-knowledge” as the ability of “knowing what you don’t know”. However, they then formulate “self-knowledge” as the ability to distinguish between answerable and unanswerable questions. Here, "unanswerable" refers to questions that cannot be answered (e.g., imaginary, subjective, or philosophical questions), while "answerable" refers to questions with a definitive answer. They then propose a dataset called "selfAware" to assess 'self-knowledge,' which contains questions with an “answerability” binary label.
>
> The terms used in [1] can cause confusion, as the “answerability” is a fixed property of the question itself (The “answerability” label in the selfAware dataset is also fixed to the question itself). So the concept “answerability” of [1] is question-dependent, not model-dependent, thus the ability to predict answerability is question-aware, not model-aware (self-aware). For example, the question “Do you like to go mountains?” is always classified as 'unanswerable' in [1], regardless of how much knowledge about mountains is stored in the answerer (e.g., 1B LM, 70B LM, or a human). This classification needs the ability of reasoning or reading comprehension, not self-awareness.
>
> Thus, [1] studies a different subject from ours, using different definitions of terms. So we had to be very careful when referencing its terminology. Nonetheless, as [1] introduces inspiring concepts, we will reference [1]. Apart from this, we effectively incorporated other works that address subjects of model-awareness (e.g., [2] [3] [4] ).
>
> ---
>
> **Reference**
>
> [1] Zhangyue Yin, et.al.,. 2023. Do Large Language Models Know What They Don’t Know?. In Findings of the ACL 2023
>
> [2]Hanning Zhang, et al. 2024. R-Tuning: Instructing Large Language Models to Say ‘I Don’t Know’. 2024 NACCL
>
> [3] Amos Azaria, et al. The internal state of an llm knows when it’s lying. arXiv preprint 2023
>
> [4] Kenneth Li, et al. Inference-time intervention: Eliciting truthful answers from a language model, NeurIPS 2024

---

> ### Author Response · Authors · 2024-11-18
> **Response to Reviewer qsGQ [3/3]**
>
> From here, we address each question, indexing the answers above.
>
> ---
>
> **[W1-1]** The problem that the paper is attempting to solve is not well justified. The concept of "Detecting hallucination before answering" appears contradictory for two primary reasons. Firstly, the term "detecting" implies that an answer already exists to be detected, which is inconsistent with the concept of doing so prior to providing an answer. From my perspective, it appears more akin to "predicting hallucination before answering".
>
> **>>** Thank you for your sharp insight. Switching to "predicting" does seem more accurate. Regarding your suggestion, for more clarity,  we changed our title into  **“Detecting Unknowns to Predict Hallucination, Before Answering: Semantic Compression Through Instruction”**
>
> ---
>
> **[W1-2]** Secondly, the necessity for "predicting" before answering is not adequately explained. Although there may be valid reasons for this, the paper fails to provide them.
>
> **>>** Addressed in Answer 3
>
> ---
>
> **[W2-1]** The paper conflates the concept of "feeling of knowing" with actual "hallucination". The authors assert that their focus is on determining when a Large Language Model (LLM) knows or does not know an answer, but they do not reference or compare their work to existing studies on the subject, such as the well-regarded "Do Large Language Models Know What They Don't Know? (Z Yin, 2023)".
>
> **>>** Addressed in Answer 5
>
> ---
>
> **[W2-2]** Furthermore, while the "feeling of knowing" is related to hallucination, the two are not identical.
>
> **>>** Addressed in Answer 1
>
> ---
>
> **[W3]** The experimental design is biased. The paper specifies that "Our baseline should predict FoK label y from question q without allowing the target LLM \theta to infer more than two steps". However, this limitation is based on the paper's specific methodology, not the task itself, and as such is unfair to the baselines
>
> **>>** Addressed in Answer 4
>
> ---
>
> **[W4]** The research on the impact of misannotation does not have a significant link with the SCAO, suggesting it could be divided into two separate papers.
>
> **>>** Addressed in Answer 2
>
> ---
>
>
> **[Q1]** "we focus on detecting when an LLM knows or does not know an answer", feeling of knowing is a different task from hallucination detection.
>
> **>>** Addressed in Answer 1
>
> ---
>
> **[Q2]**  LN068: "the effect of misannotation", what specific effect does it refer to?
>
> **>>** Addressed in Answer 2
>
> ---
>
> **[Q3]** LN080: I cannot see the connections between the contribution 1) and 2).
>
> **>>** Addressed in Answer 2.  Without 2), precisely measuring 1) is impossible.
>
> ---
>
> **[Q4]** LN105: "FoK of LLM" misses the studies about "know and don't know in LLMs". For example, "Do Large Language Models Know What They Don't Know? (Z Yin, 2023)"
>
> **>>** Addressed in Answer 5
>
> ---
>
> **[Q5]** LN312: "misannotation effect" need literature support or justification. Why is it a problem and why is it important for your topic?
>
> **>>**
> Addressed in Answer 2. It has not been explored, though it is crucial for the study of self-awareness of LLM.
>
> ---
>
> **[Q6]** LN388: "Our baseline should predict FoK label y from question q without letting the target LLM \theta infer more than two steps", why should the baseline follow the restriction of your method?
>
> **>>** Addressed in Answer 4
>
> ---
>
> **[Q7]** LN429: "impossible to address without further clue", what does it mean? Need further explanation.
>
> **>>** It means the following: without further clues such as the current year, it is impossible to determine Tampa's age. We will rephrase this to make clearer.
>
> ---
>
> **[Q8]** LN430: "we only use English questions and exclude instances with multiple labels", why?
>
> **>>** Many hallucination benchmarks and academic models are based on English, and our work is not focused on multilingual aspects. For the convenience of fellow researchers and reproducibility, we used English. Extending this to other major languages, such as Chinese, will be a task for future research.

---

> > ### Comment · Reviewer_qsGQ · 2024-11-25
> > **Thanks for clarification**
> >
> > I appreciate the authors for their comprehensive response. They have satisfactorily addressed my primary concern. Consequently, I have increased my rating from 3 to 5, given that the results are not significant on various datasets, including Mintaka, HotpotQA, and ELI5-small.

---

> ### Author Response · Authors · 2024-11-23
> **Sincere Request for Review of Our Responses and Revised Paper**
>
> Dear Reviewer qsGQ,
>
> Thank you again for your time and effort to provide your insightful feedback on our paper.
>
> We have addressed your comments and updated the draft accordingly ([click to see the pdf [link]](https://openreview.net/pdf?id=YFOg1LUGG1)). If you have any remaining questions or require further clarification, we would be happy to address them before the time window closes.
>
> Thank you so much for your time and valuable feedback!
>
> Best regards,
>
> The Authors of Paper 8793

---

> ### Author Response · Authors · 2024-11-28
>
> We are glad that our reply addressed your concerns. Your feedback helped us a lot to further clarify our paper.
>
> ---
>
> Your opinion on the experimental results is valid. However, the results still support our contribution. One of our contributions is arguing the perspective that “hallucination detection (or prediction)” is actually a collection of various tasks, and each task requires a different optimal method. We argued this by leveraging the comparison between human cognition and LLMs in Section 2. And our experimental results align with this perspective.
>
> In Section 2, we broadly categorized the causes of hallucination into two groups: 1) failure of reasoning and reading comprehension 2) knowledge recall, and we referred FoK as predicting failure in 2). However, among knowledge recall questions, there are tasks that actually require reasoning steps, and the SCAO’s performance gain appears to be not significant in this point.
>
> For more detail, we will first review the analysis of our experimental results, and then explain the concepts and contributions we propose.
>
>
> ## **1. Analysis of our experimental results: SCAO is optimal for knowledge recall tasks.**
>
> **(1) The difference in performance gains aligns with the differences in the types of questions.**
>
> As reviewer pointed out, SCAO shows significant performance gain on Explain and ParaRel, while relatively modest gain on other datasets.
> As mentioned in Section 6, we analyze that the cause of such a phenomenon mainly stems from the differences in the types of questions. ParaRel and Explain are closer to tasks that involve recalling a single piece of knowledge. In ParaRel, the recalled knowledge needs to be answered in a factoid manner, whereas in Explain, it is supposed to be answered in an open-ended manner. In contrast, datasets like HotpotQA consist of factoid questions, but answering them requires a more complex integration of diverse reasoning processes (e.g., recalling multiple knowledge and comparing). From the perspective of Section 2, these datasets may actually be closer to "after-generation" level datasets, such as MMLU.
>
> **(2) SCAO is optimal for knowledge recall tasks due to its structural characteristics.**
>
> SCAO starts from the perspective that an LLM is structurally analogous to a dense retriever. SCAO utilizes semantic compression to intensify this analogy, thereby maximizing its functionality as a retriever. For this reason, it appears to exhibit more optimal performance in straightforward knowledge retrieval tasks, such as those in Explain and ParaRel.
>
>
>
> ## **2. The experimental results support our perspective.**
>
> **(1) We need to let go of the idea that a "panacea solution” must be found for hallucination prediction.**
>
> As stated in Section 2, hallucination broadly refers to "incorrect answering," but it is actually a set of various distinct phenomena. This is because "question-answering" itself consists of various distinct tasks and scenarios, each involving different types of intelligence. In open-book settings, reasoning and reading comprehension are the primary processes, whereas in closed-book settings, knowledge recall is the primary process (and reasoning may be further engaged depending on the complexity of the question). Though we referred FoK as a subset of predicting failure in knowledge recall, there is still a portion of tasks that require reasoning steps to reach the knowledge.
>
> From this analysis, we can infer that it is difficult to find a "panacea solution" for hallucination prediction (or detection), as it is not a singular phenomenon. There will likely be an optimal solution tailored to each specific cause. In real-world scenarios, determining the optimal strategy for each question or fusing the results of several strategies would likely be the most ideal approach. As stated in Section 2, this is also how the most superior hallucination verifier in existence (our brain) operates. This represents the direction we should pursue.
>
> Most previous works have viewed "hallucination prediction (or detection)" as a singular function, and attempted to address it in a "panacea solution" manner ([1][2][3][4][5][6]). Maybe this is because most have focused on the result ("incorrect answer") itself. However, for the advance of the field of hallucination research, we argue that it is first essential to adopt and share a perspective that hallucination is a set of distinct phenomena, each involving different types of intelligence. One of our main contributions is to propose this perspective by leveraging the intensive comparison between human cognition and LLMs as in Section 2. And our experimental results align with this analysis, as SCAO has demonstrated consistent performance gains in knowledge recall tasks.

---

> ### Author Response · Authors · 2024-11-28
>
> **Reference**
>
> [1] Kurt Shuster, et al. Retrieval Augmentation Reduces Hallucination in Conversation. 2021 EMNLP
>
> [2] Hanning Zhang, et al. R-Tuning: Instructing Large Language Models to Say ‘I Don’t Know’. 2024 NACCL
>
> [3] Potsawee Manakul, et al. SelfCheckGPT: Zero-Resource Black-Box Hallucination Detection for Generative Large Language Models, 2023 EMNLP
>
> [4] Gal Yona, et al. Can Large Language Models Faithfully Express Their Intrinsic Uncertainty in Words? 2024 EMNLP
>
> [5] Sewon Min, et al. FActScore: Fine-grained Atomic Evaluation of Factual Precision in Long Form Text Generation, 2023 EMNLP
>
> [6] Jiho Kim, et al. FactKG: Fact Verification via Reasoning on Knowledge Graphs, 2023 ACL

---

> ### Author Response · Authors · 2024-12-02
> **Sincere thanks**
>
> Dear Reviewer,
>
> Thank you again for your time and effort to provide your insightful feedback on our paper.
>
> We hope and believe that our response has effectively addressed your concerns. As the discussion period is coming to an end, please let us know promptly if there are any remaining issues so that we can address them within the allotted time.
>
> Thank you so much for your time and valuable feedback!
>
> Best regards,
>
> The Authors of Paper 8793

---

### Official Review · Reviewer_A5NP · 2024-11-04

**Soundness:** 3
**Presentation:** 3
**Contribution:** 3
**Rating:** 6
**Confidence:** 3

**Summary:**

This paper proposed a method namely SCAO to detect the hallucination before full generation of LLMs. The key idea is to prompt LLMs to answer in one word. It can compress the semantic information, and assess the model's confidence. This work also introduced a method namely AME to measure mis-annotation effects in benchmarks.

**Strengths:**

1. The experiments are thorough. The method AME assessing benchmark quality is a valuable contribution.

2. The paper is well-written and easy to follow. The key concepts like FoK and SCAO are explained clearly. It is important for LLMs on hallucination detection, especially before full generation. It minimizes the computational overhead.

**Weaknesses:**

1. Although the retriever analogy is interesting, it is not very clear to me that why compressing to one word should work better than other forms of compression.

2. The gains are relatively modest in some cases. It will be great if more in-depth analysis of when and why the proposed SCAO can outperform.

**Questions:**

Did you try different forms of semantic compression rather than with one-word answers?

---

> ### Author Response · Authors · 2024-11-18
> **Response to Reviewer A5NP**
>
> **[W1]** Although the retriever analogy is interesting, it is not very clear to me that why compressing to one word should work better than other forms of compression. **[Q2]** Did you try different forms of semantic compression rather than with one-word answers?
>
> **>>**
> Thank you for having interest in our work. Exploring other types of compression methods is an area we also find highly intriguing.
> We believe compression aids FoK because it makes the structure of the LLM analogous to that of a retriever. Beyond this, we have not yet discovered other ways in which compression might benefit FoK. Various LM compression methods (e.g., xRAG [1]) have been developed recently, but these methods disrupt the alignment between the LM's output and the token space, making them unsuitable for application in our method.
>
> We propose the following additional forms of compression:
>
> **(1) Finetuning the LM Head**
>
> SCAO is a method that push LM's output close to knowledge embedded in the LM head. As mentioned in Section 3, the limitation here is that the LM head itself is not pushed toward the LM output, preventing further output. A possible alternative is to finetune the LM head along with the prompting mechanism.
>
> However, our observations showed that the performance improvement was not definitive, while the complexity of the method increased significantly. Thus, we have decided to leave this for future work.
>
> **(2) Dualization of reasoning and memory**
>
> We believe that a dualization of reasoning and knowledge storage could be one of a new possible architectures of LLM in the future; accumulating knowledge in the form of latent vector database, similar to xRAG. In such a scenario, the LM would inherently possess the structure of a retriever without requiring special compression. This could make measuring FoK through thresholding much more straightforward.
>
> ---
>
> **[W2]** The mod in some cases. It will be great if more in-depth analysis of when and why the proposed SCAO can outperform.
>
> **>>** We explained this in Section 6 (Experiments). The experimental results show that among factoid datasets, SCAO performs better on ParaRel OOD than other datasets. And in open-ended datasets, it performs better on Explain than ELI5. We propose two hypotheses to explain these findings.
>
> **(1) SCAO has an advantage in retrieving information.**
>
> Both ParaRel and Explain share a common characteristic: their questions are simple, requiring the retrieval of information about a single entity (e.g., “Please give me an explanation about Twilight”). In contrast, the questions in ELI5 and Mintaka usually require multiple processes: retrieval of information about multiple entities, reasoning and comparing with that information (e.g.,  “Who was the first wife of Queen Elizabeth II's eldest son?” “How do we know all the money the government is getting from bank settlements is going back to the people?”)
>
> The advantage of SCAO in retrieval seems to stem from its approach; it defines LLM as a retriever and maximize that characteristics by semantic compression. As discussed in the Section 2, hallucination is the result of multiple contributing factors:  the causes of hallucination are complex. failures in retrieval before generation or reasoning errors after generation. There are optimal strategies for detecting failures at each stage, and SCAO can be considered an optimal strategy for retrieval.
>
> **(2)SCAO is advantageous in out-of-distribution (OOD) scenarios.**
>
> ParaRel OOD is a dataset where the train set and test set are separated. We have analyzed in the Appendix B why confidence-based methods could perform better in OOD scenarios.
>
> ---
>
> **Reference**
>
> [1] Xin Cheng, et.al.,  xrag: Extreme context compression for retrieval-augmented generation with one token, 2024 NeurIPS

---

> ### Author Response · Authors · 2024-11-23
> **Sincere Request for Review of Our Responses and Revised Paper**
>
> Dear Reviewer A5NP,
>
> Thank you again for your time and effort to provide your insightful feedback on our paper.
>
> We have addressed your comments and updated the draft accordingly ([click to see the pdf [link]](https://openreview.net/pdf?id=YFOg1LUGG1)). If you have any remaining questions or require further clarification, we would be happy to address them before the time window closes.
>
> Thank you so much for your time and valuable feedback!
>
> Best regards,
>
> The Authors of Paper 8793

---

> ### Author Response · Authors · 2024-12-02
> **Sincere thanks**
>
> Dear Reviewer,
>
> Thank you again for your time and effort to provide your insightful feedback on our paper.
>
> We hope and believe that our response has effectively addressed your concerns. As the discussion period is coming to an end, please let us know promptly if there are any remaining issues so that we can address them within the allotted time.
>
> Thank you so much for your time and valuable feedback!
>
> Best regards,
>
> The Authors of Paper 8793

---

### Official Review · Reviewer_RzuC · 2024-11-04

**Soundness:** 3
**Presentation:** 3
**Contribution:** 2
**Rating:** 6
**Confidence:** 3

**Summary:**

This work focuses on addressing hallucinations in large language models, by estimating to what extent the model has relevant knowledge— relying on the fact that model hallucinations primarily arise when the model is making extrapolations. The work proposes a method “semantic compression via trying to answer in one word” to estimate to what extent the model has relevant knowledge or not (FOK).

**Strengths:**

(1) The paper is well-written and clear.

(2) The experimental methodology is well laid out and the rationale behind various decisions is clearly explained in the manuscript.

(3) This work joins a growing line of research in estimating model confidence and knowledge, which is an interesting research direction.

**Weaknesses:**

(1) The evaluation is mostly done focusing on QA tasks where there is a concept of a right answer or a set of right answers. However, many times hallucinations can happen in more open-ended settings, like with the prompt “Tell me a Biography of Barack Obama” (see Min et al., 2023). Models may interleave true facts and false facts when generating a response, how can this method be extended to these typical long-form settings?

(2) It would be great to include more analysis of when this method works and when it doesnt, for example where in the model response the correct answer string occurs when making the FOK dataset.

Min, Sewon, et al. "Factscore: Fine-grained atomic evaluation of factual precision in long form text generation." arXiv preprint arXiv:2305.14251 (2023)

**Questions:**

Could the authors explain how SCAO works when a question is not answerable by a single word?

---

> ### Author Response · Authors · 2024-11-18
> **Response to Reviewer RzuC**
>
> **[W1]** The evaluation is mostly done focusing on QA tasks where there is a concept of a right answer or a set of right answers. However, many times hallucinations can happen in more open-ended settings, like with the prompt “Tell me a Biography of Barack Obama” (see Min et al., 2023). Models may interleave true facts and false facts when generating a response, how can this method be extended to these typical long-form settings?
>
> **>> Our paper already contains open-ended settings like FActScore**
>
> Thank you for your interest in our work and for raising such thoughtful questions. However, we also had an interest in this subject and already addressed this aspect in Section 6.3. Specifically, we conducted experiments on open-ended datasets, including ELI5 and Explain. Explain is a newly proposed dataset that uses an experimental setup that is nearly identical to what was used in FActScore [1] you referred, but ours extends and builds upon [1] further.
>
> In [1], a small dataset is created by appending prompts like “Tell me a bio of  <entity>”  to person names sourced from Wikipedia, prompting descriptive answers. However, a limitation of this dataset is that its subjects are restricted to person names and it includes only 500 entries. To address this, we developed Explain. In Explain, the subjects are more general, covering people, history, buildings, culture, and more, with the dataset size expanded to about 15,000 entries. The prompt is "Please give me an explanation about <entity>", which is nearly identical to FActScore.
>
> Therefore, the FoK performance in the open-ended scenario (which you are curious about) has already been evaluated, and SCAO was observed to achieve advanced performance.
>
> ---
>
> **[W2]** It would be great to include more analysis of when this method works and when it doesnt, for example where in the model response the correct answer string occurs when making the FOK dataset. **[Q1]** Could the authors explain how SCAO works when a question is not answerable by a single word?
>
> **>> Analysis on how SCAO reacts to open-ended question**
>
> The reviewer has provided an excellent question that greatly aids in understanding our work. We will include the section with table and figure to address this question in Appendix C.4.
>
> Since Explain is an open-ended dataset whose question is not answerable by a single word, we will provide an analysis of the first token candidate for both one-word compressed and non-compressed cases within this dataset.
>
> First, in non-compressed cases (queried with a normal prompt), the following patterns are frequently observed: (1) The response often starts by repeating the entity name mentioned in the query. (2) The response begins with grammatical function words such as "The" or "A". In other words, the model tends to take the easy path. As a result, the probability of the initial token is generally inflated, regardless of whether the model truly knows the subject.
>
> On the other hand, when prompted to answer with a one-word response, the first token often corresponds to the initial token of a word encapsulating the entity's characteristics. For example, in response to the question "Please give me an explanation about 'Breaking Dawn'.", the first candidate token was "Tw" (the first token of "Twilight"). In other words, with one-word prompting, the model shows a stronger tendency to retrieve its own knowledge related to the entity.
>
> This trend is also reflected statistically. Among the 2152 test samples in the Explain dataset, the case that the  top-1 candidate of the first token of the response being a component of the entity is 84.5% for normal prompting, significantly outpacing the 12.1% for one-word prompting. Similarly, the first token being "the" occurred in 17.8% of normal prompting cases, compared to just 0.02% for one-word prompting.
>
> ---
>
> **Reference**
> [1] Sewon Min, et. al., 2023. FActScore: Fine-grained Atomic Evaluation of Factual Precision in Long Form Text Generation. 2023 EMNLP

---

> ### Author Response · Authors · 2024-11-23
> **Sincere Request for Review of Our Responses and Revised Paper**
>
> Dear Reviewer RzuC,
>
> Thank you again for your time and effort to provide your insightful feedback on our paper.
>
> We have addressed your comments and updated the draft accordingly ([click to see the pdf [link]](https://openreview.net/pdf?id=YFOg1LUGG1)). If you have any remaining questions or require further clarification, we would be happy to address them before the time window closes.
>
> Thank you so much for your time and valuable feedback!
>
> Best regards,
>
> The Authors of Paper 8793

---

> ### Author Response · Authors · 2024-12-02
> **Sincere thanks**
>
> Dear Reviewer,
>
> Thank you again for your time and effort to provide your insightful feedback on our paper.
>
> We hope and believe that our response has effectively addressed your concerns. As the discussion period is coming to an end, please let us know promptly if there are any remaining issues so that we can address them within the allotted time.
>
> Thank you so much for your time and valuable feedback!
>
> Best regards,
>
> The Authors of Paper 8793

---

### Official Review · Reviewer_p2Rf · 2024-11-06

**Soundness:** 2
**Presentation:** 4
**Contribution:** 3
**Rating:** 5
**Confidence:** 5

**Summary:**

This paper introduces a method for detecting when an LLM knows the answer to a question *before* generating it. This method relies on probing the last token logits and hidden state obtained via a special "answer-in-one-word" prompt attached to the original query. The paper also introduces a procedure for identifying datasets with misannotations and excludes these from the analysis. Experimental results on entity-centric datasets suggest that the proposed method significantly improves the accuracy of detecting questions where the LLM is likely to provide a correct answer.

**Strengths:**

- The problem formulation of detecting a hallucination *without* generating an answer is quite interesting and potentially useful.
- The main idea of representing the query via a "answer-in-one-word" prompt is intuitive and interesting.
- Several baselines are evaluated and the proposed methods shows notable gains in accuracy of detecting hallucinated answers.

**Weaknesses:**

- The AME analysis for detecting misannotations is unconvincing. The main idea is that if a classifier can detect hallucinated answers based on the question alone, this points to some issue with the question. But such a classifier might be simply detecting difficult questions or questions about tail knowledge. In this case, it is not clear why datasets with such questions should be excluded? Moreover there isn't any further evidence provided that the datasets excluded based on AME were indeed problematic.
- Max token length is set to 50 when generating answers for the long-form QA datasets -- this seems very short for a dataset like ELI5. Further, it is not clear why a new dataset like Explain was used rather than other existing long-form QA datasets (e.g., ASQA).
- The main motivation for detecting a hallucination before generating is to save the cost of decoding. However, most of the empirical analysis in this paper is on datasets where the answers are short entities. In such a case, appending the answer-in-one-word prompt is perhaps equally expensive as decoding the answer. Hence, the paper would benefit from evaluating on more realistic setups where the answers are longer. (There are some results in Table 3, but these are not very convincing).

**Questions:**

- How does SCAO compare to R-tuning on the same datasets as used in the latter paper?

---

> ### Author Response · Authors · 2024-11-18
> **Response to Reviewer p2Rf [1/2]**
>
> We truly appreciate your deep understanding of our paper. We address the key issue raised by the reviewer in the comments below:
>
> ---
> **[W1]**
> The AME analysis for detecting misannotations is unconvincing. The main idea is that if a classifier can detect hallucinated answers based on the question alone, this points to some issue with the question. But such a classifier might be simply detecting difficult questions or questions about tail knowledge. In this case, it is not clear why datasets with such questions should be excluded?
>
> **>>  AME is to filter out the effect of question-dependent hallucination**
>
> We appreciate the reviewer for pointing out a sharp aspect that clarifies the understanding. The confusion between “hallucination detection” and “FoK” seems to have prompted these questions. We will provide further clarification to address this.
>
> FoK is the ability to be aware of whether LLM possesses specific knowledge. To evaluate this, what we can explicitly measure is the amount of incorrect answers (hallucinations, denote H). Causes of incorrect answers can be naively divided into two: question-dependent and model-dependent. (1) Question-dependent: This occurs when the question is difficult or unanswerable (denote Q). (2) Model-dependent: This happens when the model lacks the necessary knowledge to answer the question (denote M).
>
> When H=Q+M, FoK refers to predicting the portion of M. Since H is measurable, we should measure Q to accurately determine M. Measuring Q is exactly what AME does, as it predicts H based solely on question-dependent factors.
>
> As you mentioned, predicting question difficulty (measuring Q) can indeed help clarify the H. However, as our work focuses on FoK and experiment the FoK performance of methods, we should filter out the portion of Q.
>
> (To emphasize this aspect, we redefine AME as **AQE (approximate question-dependency effect)** in the revised paper)
>
> ---
>
> **[W1-2]**
>  Moreover there isn't any further evidence provided that the datasets excluded based on AME were indeed problematic.
>
> **>> Datasets with high AME exhibits failure in covering multiple labels**
>
>  We described the characteristics and examples of datasets with high AME in the introduction and Section 5. These datasets contains questions nearly impossible to answer correctly. This arises from the failure to properly constrain the one-to-many mapping between question and answer, which can be considered as misannotation.
>
> The SimpleQA, which recorded the highest AME (82%), contains questions like “What is a Western genre on Netflix?”. Though there are countless Western genre movies on Netflix, this dataset provides only one label ("Rawhide"). Even if the LM possesses extensive knowledge about Netflix, any answer other than "Rawhide" will be labeled as incorrect. As there are multiple similar types of questions (e.g., “What is a romance genre on Netflix?”, “What is a action genre on Netflix?”), this can cause the bias that any question on Netfilx is paired with only a negative FoK label. This bias makes the FoK dataset question-dependent, raising the AME score.
>
> In contrast, Mintaka, which has a lower AME (60%), contains questions with detailed information to ensure that each question has only one label (e.g., “Who was the first wife of Queen Elizabeth II's eldest son?”). Such questions may appear as detailed tail questions but help prevent the misannotation effect, resulting in lower AME.
>
> ---
>
> **[W2-1]** Max token length is set to 50 when generating answers for the long-form QA datasets -- this seems very short for a dataset like ELI5.
>
> **>> We add experiment with max length 256**
>
> Thank you for the constructive feedback; your points are very persuasive. We add experimental results with a longer max token length (256) in Appendix C.3 of revised paper. As the response length increased, the True rate of ELI5 improved. And the overall tendency of the FoK results remains the same: SCAO clearly outperforms in Explain, while there is no significant difference in ELI5.

---

> ### Author Response · Authors · 2024-11-18
> **Response to Reviewer p2Rf [2/2]**
>
> **[W2-2]**
> Further, it is not clear why a new dataset like Explain was used rather than other existing long-form QA datasets (e.g., ASQA)., **[W3-1]** The main motivation for detecting a hallucination before generating is to save the cost of decoding. However, most of the empirical analysis in this paper is on datasets where the answers are short entities. In such a case, appending the answer-in-one-word prompt is perhaps equally expensive as decoding the answer. Hence, the paper would benefit from evaluating on more realistic setups where the answers are longer. (There are some results in Table 3, but these are not very convincing).
>
> **>> Our datasets already consist of factoid question with long-form answering**
>
> Thank you for asking questions that help us refine and use the terminology in our research more accurately. There seems to be some confusion in the terminology in our paper (long vs short form, and open-ended vs factoid), so we will correct this. Our datasets, such as Mintaka, ParaRel, Halueval, and HotpotQA, are factoid question datasets with long-form answer options, which are the same as the ASQA you mentioned.
>
> To explain in more detail, there are two dimensions that distinguish between "long" and "short": the label and the answer. (1) label dimension: This refers to whether the label of a question is short or long. A factoid question is one where the label is a short entity name, while an open-ended question has a label that is lengthy and has various versions. (2) question dimension: Independent of the label's length, the answer leading to it can be lengthy and complex. This is referred to as long-form question answering, while answering with only a few words is short-form.
>
> Using these terms, ASQA is a long-form factoid question answering dataset. Label is short entity, and evaluation is conducted through ROUGE or string match. One of the example question  "When was the first Apple i phone made?" is akin to questions from our datasets such as Mintak or ParaRel.
>
> From that perspective, our datasets are also long-form QA with factoid questions. We allowed responses within a maximum token limit of 50. In this regard, our study has applicability.
>
> **>> The setup of Explain is adopted from the existing work of FActScore.**
>
> The setup of the dataset Explain we proposed is an extended and refined version of the already existing and verified open-ended long-form dataset in the work of FActScore [1]. In [1], a small dataset is devised to test fact-checking methods for long-form QA. This dataset is created by appending prompts like “Tell me a bio of  <entity>”  to person names sourced from Wikipedia.
>
> However, its subjects are limited to only person names, and it includes only 500 entries. To address this, we developed Explain. In Explain, the subjects are more general, covering people, history, buildings, culture, and more (the entities from Mintaka), with the dataset size expanded to about 15,000 entries. The prompt is "Please give me an explanation about <entity>", which follows the motivation of the dataset in FactScore.
>
>
> ---
>
> **[Q1]** How does SCAO compare to R-tuning on the same datasets as used in the latter paper?
>
> **>>**
> The detailed application of R-tuning is described in Section 6.1 and Appendix D.2.2 (revised version).
> While the original work conducted R-tuning with the data form of “question + answer + sure/unsure expression” (note as R-tuning in the Table 2), we also train with the form without answer “question + sure/unsure expression” (note as R-tuning (q only), as we assume a before-generation FoK scenario.
> Additionally, for a more fair comparison, distinct from the original work, we train a separate LoRA adapter as a $\phi$, then let $\phi$ predict the FoK label of the body LLM $\theta$. The original work directly trains $\theta$ itself as a $\phi$. This modification is to address the catastrophic forgetting problem \citep{jang2021towards} during training $\theta$ directly. We observe that the True rate decreases by 13.2\%p after R-tuning, seriously undermining the justification of the method. We train for one epoch with a global batch size of 16 and a learning rate of 1e-5, as it is reported that a small batch size is better for R-tuning.
>
> ---
> **Reference**
>
> [1] Sewon Min, et. al., 2023. FActScore: Fine-grained Atomic Evaluation of Factual Precision in Long Form Text Generation. 2023 EMNLP

---

> ### Author Response · Authors · 2024-11-23
> **Sincere Request for Review of Our Responses and Revised Paper**
>
> Dear Reviewer p2Rf,
>
> Thank you again for your time and effort to provide your insightful feedback on our paper.
>
> We have addressed your comments and updated the draft accordingly ([click to see the pdf [link]](https://openreview.net/pdf?id=YFOg1LUGG1)). If you have any remaining questions or require further clarification, we would be happy to address them before the time window closes.
>
> Thank you so much for your time and valuable feedback!
>
> Best regards,
>
> The Authors of Paper 8793

---

> > ### Comment · Reviewer_p2Rf · 2024-11-25
> >
> > Thank you to the authors for the revisions and detailed responses. However, I will keep my original score.
> >
> > - Despite the explanation, the AQE analysis still remains unconvincing. The classifier trained in the paper to predict incorrect answers, still relies on supervision derived from the model’s predictions. Hence, the subset it predicts is not solely “question-dependent” — it also relies on the labels about which questions the model predicted incorrectly, and hence indirectly depends on the model’s knowledge.
> > - The qualitative analysis suggests that the actual purpose that AQE serves is to determine which datasets have a one-to-many mapping between questions and answers. This can probably be done more directly by simply asking an LLM to classify questions on this aspect.
> > - Thanks for clarifying on the question vs label dimensions of “long-form” QA. However, since a key motivation of the paper is based on saving the cost of generating from an LLM (by detecting FoK). Hence, I still feel the paper should have evaluated on more datasets where longer generations are necessary (like ELI5). Factoid datasets, as the response points out, can always be answered using a small number of tokens.
> > - The additional results with token length of 256 are a nice addition.

---

> > > ### Author Response · Authors · 2024-11-25
> > > **Second response to Reviewer p2RF _3**
> > >
> > > **Reference**
> > >
> > > [1] Jason Wei, Chain-of-Thought Prompting Elicits Reasoning in Large Language Models, 2022 NeurIPS
> > >
> > > [2] Ivan Stelmakh, et al. Asqa: Factoid questions meet long-form answers, 2023.
> > >
> > > [3] Tom Kwiatkowski, et al. Natural questions: a benchmark for question answering research. 2019 ACL
> > >
> > > [4] Pranav Rajpurkar, et al. SQuAD: 100,000+ Questions for Machine Comprehension of Text. 2016 EMNLP
> > >
> > > [5] Stephanie Lin, et al. TruthfulQA: Measuring How Models Mimic Human Falsehoods. 2022 ACL
> > >
> > > [6] James Thorne, et al. FEVER: a Large-scale Dataset for Fact Extraction and VERification. 2018 NAACL
> > >
> > > [7] Dan Hendrycks, et al. Measuring Massive Multitask Language Understanding.. 	2021ICLR
> > >
> > > [8] Rachneet Sachdeva, et al. Localizing and Mitigating Errors in Long-form Question Answering, 2024
> > >
> > > [9] Sewon Min, et al. FActScore: Fine-grained Atomic Evaluation of Factual Precision in Long Form Text Generation, 2023 EMNLP
> > >
> > > [10] Hanning Zhang, et al. R-Tuning: Instructing Large Language Models to Say ‘I Don’t Know’. 2024 NACCL
> > >
> > > [11] Gal Yona, et al. Can Large Language Models Faithfully Express Their Intrinsic Uncertainty in Words? 2024 EMNLP
> > >
> > > [12] Kenneth Li, et al. Inference-Time Intervention: Eliciting Truthful Answers from a Language Model. 2023 NeurIPS
> > >
> > > [13] Kalpesh Krishna, et al. Hurdles to Progress in Long-form Question Answering. 2021 NAACL

---

> ### Author Response · Authors · 2024-11-25
> **Second response to Reviewer p2RF _1**
>
> Thank you so much for your insightful feedback. Your feedback deepens our understanding of our own work.
>
> ---
>
> **[Q1]**. Despite the explanation, the AQE analysis still remains unconvincing. The classifier trained in the paper to predict incorrect answers, still relies on supervision derived from the model’s predictions. Hence, the subset it predicts is not solely “question-dependent” — it also relies on the labels about which questions the model predicted incorrectly, and hence indirectly depends on the model’s knowledge.
>
> **>> AQE is a tool for measuring the effect of “being aware of the model's inner state”**
>
> The reviewer's opinion is very insightful but a little confused.
>
> As the reviewer mentioned, since the classifier predicts whether the model gives an incorrect answer, we can say that the prediction reflects the model's knowledge. "Whether the model gives an incorrect answer" itself (denoted $H$) is the model's knowledge. And the portion of $H$ (model's knowledge) that can be inferred solely dependent on the question, was defined as $Q$ (question dependency).
>
> For example, if a model lacks knowledge in the science category, the AQE classifier can infer whether the model fails to answer (model's knowledge) solely by categorizing the subject of the questions. In this case, we can say the AQE classifier predicted the model's knowledge solely dependent on the question.
>
> In contrast, some aspects of the model's knowledge can only be inferred dependent on the model's inner state (e.g., hidden state, confidence). We defined this portion as $M$ (model dependency).
>
> For example, if a model lacks knowledge in the science category, but it only knows the fact that the Earth is round. When the model is asked "What shape is the Earth?", it can generate the answer "It is round" correctly with a high confidence score. In this case, the question-dependent AQE classifier will fail to predict the correctness, while the model-dependent classifier will succeed.
>
> In short, among the model’s knowledge ($H$), a portion that can be predicted solely dependent on the question is $Q$, while the rest (supposed to be a portion that needs the model's inner state) is $M$. And AQE is measuring $Q$ to clarify the FoK classifier's ability to be aware of model's inner-state.
>
> In addition, the above examples indicate two things. 1) Question-dependent prediction can also provide guessing for FoK prediction. 2) However, relying on such guessing is akin to absorbing bias (sometimes only applicable to a certain dataset), which limits the resolution and generalization, ultimately disrupting more precise FoK. This is because the model is diverse and dynamically evolves, making it insufficient to rely solely on the bias from questions. Therefore, measuring model-awareness is crucial for developing an accurate FoK method.
>
> ---
>
> **[Q2]** The qualitative analysis suggests that the actual purpose that AQE serves is to determine which datasets have a one-to-many mapping between questions and answers. This can probably be done more directly by simply asking an LLM to classify questions on this aspect.
>
> **>>**
>
> The reviewer pointed out a good point. This can be explained in two ways.
>
> **1. One-to-many mapping is one of the main causes**
>
> The failure of covering one-to-many mapping between a question and answers is a key case where question-dependency occurs. However, question-dependency may arise through various latent pathways that we do not yet fully understand. AQE is supposed to comprehensively capture the effect of those latent pathways.
>
> For example, if the failure to cover one-to-many mapping was the only cause of question-dependency, the AQE for Explain and ELI5l should be close to 0.5, as they utilize G-eval that is capable of flexibly covering one-to-many mapping. However, both still exhibit a fairly high AQE. The AQE classifier might have identified certain question patterns that are yet difficult to discern through human intuition. Whatever it is, it is a pattern that can be identified without information about the model's inner state. Such cases cannot be identified by asking an LLM but can be captured by AQE.
>
> **2. Asking an LLM has the following limitations:**
>
> **1)** Its results are dependent on the LLM's reasoning ability and knowledge capacity, which can vary significantly across different LLMs. **2)** The answer to whether a question is question-dependent will largely depend on the prompt as well. What is the best choice for the prompt and LLM?  The criteria for this are likely to be highly controversial and involve heuristics. Therefore, asking LLMs is at high risk of being criticized as unreliable. In contrast, AQE is more reliable, as the measured values are consistent as long as the target dataset and model are fixed. **3)** Asking high-performing LLMs (e.g., GPT-4) is costly, whereas AQE requires very minimal resources.

---

> ### Author Response · Authors · 2024-11-25
> **Second response to Reviewer p2RF _2**
>
> **[Q3]** Thanks for clarifying the question vs label dimensions of “long-form” QA. However, since a key motivation of the paper is based on saving the cost of generating from an LLM (by detecting FoK). Hence, I still feel the paper should have evaluated on more datasets where longer generations are necessary (like ELI5). Factoid datasets, as the response points out, can always be answered using a small number of tokens.
>
> **>>**
>
>  The reviewer's comment is absolutely correct. It directly points to the direction in which research on FoK should progress. However, for now, it can be explained as follows.
>
> **1. Factoid dataset is also a sufficiently meaningful long-form question-answering setting.**
>
> It is because when we ask factoid questions to LLM services like ChatGPT, we do not simply expect a one-word answer. According to the Chain of Thought theory [1], as the response length increases, the accuracy of the answer may also improve. For this reason, most of the datasets we know as LFQA datasets (e.g., ASQA[2], NQ[3], SQuAD[4], TruthfulQA[5]) are factoid datasets.
>
> **2. We reviewed as many open-ended datasets as our best, and the only valid ones are ELI5 and Explain.**
>
> We tried to review as many open-ended datasets as possible. However, most of the datasets known as long-form question datasets  (e.g., ASQA, NQ, SQuAD, TruthfulQA) turned out to be factoid datasets. Or they are reading comprehension tasks (e.g., FEVER[6]). Or they were datasets designed to measure multi-step reasoning abilities (e.g., MMLU[7]), which are clearly categorized to the after-generation process.
>
> Among more suitable datasets, HaluQuestQA[8] contains open-ended long-form questions similar to ELI5. However, as it contains only 898 questions, it is not valid for the FoK test.
>
> A work of FActScore[9] contains an open-ended long-form dataset that can evaluate the ability to explain certain knowledge, but it also contains only 500 questions, with topics limited to only person names. Therefore, we modified it to create a new valid open-ended dataset Explain.
>
> As you can see, we made every effort to conduct sufficient experiments on open-ended datasets.
>
> **3. One of the main motivations of our work is to test the feasibility of the concept "FoK before generation", and factoid questions are more suitable for this.**
>
> One of the purposes of our work is to explore the concept of "FoK before generation" itself. In contrast to the previous works that mainly focused on hallucination detection “after generation”, we focused on the hypothesis that it is also possible before generation by utilizing information from the model's inner state. This is a valuable topic in itself that is worth exploring. In exploring this topic, utilizing factoid datasets is necessary for the following reasons.
>
> **1)** Previous studies on “hallucination detection after generation” (e.g., R-tuning[10], [11], [12]) also utilize factoid datasets such as TruthfulQA, Mintaka, and ParaRel. Conducting experiments on factoid datasets allows us to demonstrate our feasibility in continuity with previous research.
>
> **2)** An open-ended dataset has an inherent limitation that determining the correctness of an answer is non-trivial. One of the reasons we did not present ELI5 as a main experimental setting is also that the evaluation of the correctness of answers in ELI5 remains unclear, and this can make FoK itself unclear.
>
> This limitation has been consistently highlighted in other works as well ([2] [13] [9]). Many works are exploring how to measure this, but it seems that consensus has not been reached yet. Due to the ambiguity in defining the correctness of an answer, evaluating the performance of predicting this correctness (FoK) can also be unreliable. Therefore, focusing on open-ended datasets may make it challenging to demonstrate the feasibility of FoK before generation. A relatively reliable setting is one that contains only factual knowledge, such as Explain. For this reason, we created and conducted experiments in Explain.
>
> Of course, open-ended questions are a highly important challenge and a key direction for future research. However, this must be accompanied by advancements in the field of long-form QA itself.

---

> ### Author Response · Authors · 2024-12-02
> **Sincere thanks**
>
> Dear Reviewer,
>
> Thank you again for your time and effort to provide your insightful feedback on our paper.
>
> We hope and believe that our response has effectively addressed your concerns. As the discussion period is coming to an end, please let us know promptly if there are any remaining issues so that we can address them within the allotted time.
>
> Thank you so much for your time and valuable feedback!
>
> Best regards,
>
> The Authors of Paper 8793

---

### Author Response · Authors · 2024-11-18
**Global Rebuttal**

We first thank all reviewers for their thoughtful feedback on our work. We believe that constructive suggestions from all reviewers, such as the clear use of terminology between hallucination detection and FoK (Feeling of Knowing), can significantly enhance the clarity and advancement of our paper.

We have incorporated this feedback to update our paper, and the key updates are summarized below.

---
## 1. Clarification of the term  "hallucination detection" and "FoK"
As reviewer qsGQ pointed out, hallucination detection and FoK are distinct concepts. While we have already explained this in Sections 1 and 2, there still seems to be room for more emphasis. Therefore, we have revised our paper to clarify this further.

**Definition of "hallucination detection" and "FoK"**

Here, we review how the definitions of "hallucination detection" and "FoK" differ. While hallucination indicates general incorrect answering, its causes can be categorized into various factors. In open-book tasks, hallucination is primarily caused by issues with reasoning and reading comprehension, whereas in closed-book tasks, it is mainly due to a lack of certain knowledge. Additionally, the question itself being unanswerable or vague can also be a cause.

Among these, FoK refers to detecting the lack of certain knowledge in closed-book scenarios. Therefore, FoK can be considered a subset of hallucination detection. Further, our work focuses on the scenario of detecting before generation.

Despite these differences, we used "hallucination detection" in the title for the following reasons: 1) In common service environment like chatGPT, it is well-known that a major cause is when the model is asked questions in subjects where it lacks pre-trained knowledge. LLMs lack the ability to say "I don't know" when they encounter unknowns, leading to a hallucinated answer instead. 2) Previous works have often regarded these two concepts as identical [1], without providing a separate definition of FoK. In connection with prior research, we chose this expression.

However, this can cause confusion. Thus, to avoid confusion, we thoroughly emphasized the distinction between the two concepts in the paper. Nevertheless, as there is still potential for misunderstanding, we update the paper as follows.

**Key updates in our paper**

1) We change the title of the paper as follows for clarity.

*Detecting Unknowns to Predict Hallucination, Before Answering: Semantic Compression Through Instruction*

2) We revised Sections 1 and 2 to further clarify the distinctions between the two concepts, "hallucination detection" and "FoK."

3) Our proposed metric for assessing datasets, AME, is designed to clarify the evaluation of FoK itself rather than hallucination detection. Accordingly, we revised the explanation of AME in Section 5 to better reflect this purpose.

---
## 2. Redefining AME (approximate misannotation effect) to AQE (approximate question-dependency effect)

Regarding the review of p2RF, we redefined AME as AQE and revised its explanation in Section 5 to emphasize the following aspects:
1) We emphasize that AQE not only measures the effect of misannotated questions but also captures broader question-dependency in the FoK datasets.
2) We clarify that AQE is a metric designed to evaluate the model-awareness of FoK methods, rather than being solely focused on hallucination detection.

The revised explanation of AQE is as follows. (identical to the paper):

**Explanation on AQE**

For a more precise evaluation of FoK, we provide a metric that can assess whether a dataset is fit for evaluating FoK. While FoK is the ability to be aware of whether LLM possesses specific knowledge, what we can explicitly measure is only the amount of incorrect answers (hallucinations, denote $H = \sum_i  1 (y_i = False)$), which is not enough to measure self-awareness.

Causes of incorrect answers can be naively divided into two: question-dependent and model-dependent. **(1) Question-dependent:** This occurs when the question is difficult or unanswerable (denote $Q$). **(2) Model-dependent:** This happens when the model lacks the necessary knowledge to answer the question (denote $M$). When $H=Q+M$, FoK refers to predicting the portion of $M$. Since $H$ is measurable, we should measure $Q$ to accurately determine $M$. But measuring $Q$ directly is non-trivial, as it is for $M$.

AQE is devised to approximate the $Q$.  $Q$ is defined as the portion of incorrect answers ($H$) that can be predicted solely with the properties of the question, independent of $\theta$. To fit this definition, we train and test a model $\phi$ to predict $y=False$ case using only the question as input. This accuracy of $\phi$ is the AME score. The closer AME is to 1, the lower the model-dependency of the dataset (paired with a certain model), making it unsuitable for measuring FoK.

**Reference**
[1]Hanning Zhang, et al. R-Tuning: Instructing Large Language Models to Say ‘I Don’t Know’.  2024 NACCL

---

### Author Response · Authors · 2024-11-24

Dear Reviewers.

For a more convenient review of the revised paper, we provide a version where the major edits compared to the first submitted draft are highlighted in **blue** ([click to see the pdf [link]](https://openreview.net/pdf?id=YFOg1LUGG1)). After the discussion period ends, we will remove the highlight for the final version submission.

Through the rebuttal and updates of our paper, we believe we have addressed all of the reviewers' concerns. This enabled us to write a more refined and comprehensive draft. We deeply appreciate the reviewers' insightful feedback.

As the end of the discussion period approaches, please feel free to raise any remaining questions or suggestions.

Thank you so much for your time and valuable feedback!

Best regards,

The Authors of Paper 8793

---

### Author Response · Authors · 2024-11-28

Dear Reviewers.

As the PDF revision period came to an end, we convert the version with blue-highlighted revisions back to a version without highlights ([click to see the pdf[link]](https://openreview.net/pdf?id=YFOg1LUGG1))).
The blue-highlighted version is still accessible in the revision history (revised version on 25 Nov 2024 ([click to see the pdf [link]](https://openreview.net/notes/edits/attachment?id=jis0eX23UI&name=pdf))).

If you have any remaining concern, we would be happy to address any further questions/suggestions that might come up until the end of the discussion period.

Thank you so much for your time and valuable feedback!

Best regards,

The Authors of Paper 8793

---

### Meta-Review · Area_Chair_BkeT · 2024-12-20

**Metareview:**

This paper presents an approach to assess whether an LLM knows the answer to a question prior to generating a response. The approach involves probing the LLM's final token logits and hidden states using a specialized "answer-in-one-word" prompt. Additionally, the paper proposes a process for identifying and excluding datasets with misannotations from the analysis.

While we appreciate the authors’ responses during the rebuttal phase, several key concerns raised by the reviewers remain unresolved:

- The experimental findings lack sufficient significance to bolster the paper’s contributions. The results across datasets such as Mintaka, HotpotQA, and ELI5-small are not particularly compelling.
- The experiments focus solely on factoid QA, which is insufficient to justify the proposed approach. As highlighted by one reviewer, a core motivation of the paper is to reduce the cost of generating responses from an LLM. Consequently, it is crucial to evaluate the approach on datasets that require longer generations.

These unresolved issues limit the overall value of the work. As a result, we agree that the current submission does not meet the ICLR standard. We hope these reviews provide useful feedback for the authors’ future revisions.

**Additional Comments On Reviewer Discussion:**

The authors have addressed most of the points. The remained unsolved points have been listed in the meta-review.

Specifically, even for the second unsolved point, the authors have provided new results trying to justify their claim. While I would agree with Reviewer p2Rf that more experiments on long text generation is necessary. Therefore, finally I recommend a reject with a "bump-up" confidence.

---

### Decision · Program_Chairs · 2025-01-22

Reject